# Impact of the Variability in Vertical Separation between Biomass-Burning Aerosols and Marine Stratocumulus on Cloud Microphysical Properties over the Southeast Atlantic

Siddhant Gupta[1,2], Greg M. McFarquhar[1,2], Joseph R. O'Brien[3], David J. Delene[3], Michael R. Poellot[3], Amie Dobracki[4], James R. Podolske[5], Jens Redemann[2], Samuel E. LeBlanc[5,6], Michal Segal-Rozenhaimer[5,6,7] and Kristina Pistone[5,6]

[1]Cooperative Institute for Mesoscale Meteorological Studies, University of Oklahoma, Norman, OK, USA
[2]School of Meteorology, University of Oklahoma, Norman, OK, USA
[3]Department of Atmospheric Sciences, University of North Dakota, Grand Forks, ND, USA
[4]Department of Atmospheric Sciences, Rosenstiel School of Marine and Atmospheric Science, University of Miami, Miami, FL, USA
[5]NASA Ames Research Center, Moffett Field, CA, USA
[6]Bay Area Environmental Research Institute, Moffett Field, CA, USA
[7]Department of Geophysics and Planetary Sciences, Porter School of Environmental and Earth Sciences, Tel Aviv University, Tel Aviv, Israel

*Correspondence to*: Siddhant Gupta (sid@ou.edu)

**Abstract.** Marine stratocumulus cloud properties over the southeast Atlantic Ocean are impacted by contact between above-cloud biomass-burning aerosols and cloud tops. Different vertical separations (0 to 2000 m) between the aerosol layer and cloud tops were observed on six research flights in September 2016 during the NASA ObseRvations of Aerosols above CLouds and their intEractionS (ORACLES) field campaign. There were 30 *contact* profiles where an aerosol layer with aerosol concentration ($N_a$) > 500 cm$^{-3}$ was within 100 m of cloud tops, and 41 *separated* profiles where the aerosol layer with $N_a$ > 500 cm$^{-3}$ was located more than 100 m above cloud tops. For *contact* profiles, the average cloud droplet concentration ($N_c$) in the cloud layer was up to 68 cm$^{-3}$ higher, the effective radius ($R_e$) up to 1.3 µm lower and the liquid water content (LWC) within 0.01 g m$^{-3}$ compared to *separated* profiles. Free tropospheric humidity was higher in the presence of biomass-burning aerosols and *contact* profiles had a smaller decrease in humidity (and positive buoyancy) across cloud tops with higher median above-cloud $N_a$ (895 cm$^{-3}$) compared to *separated* profiles (30 cm$^{-3}$). Due to droplet evaporation from entrainment mixing of warm, dry free tropospheric air into the clouds, the median $N_c$ and LWC for *contact* profiles decreased with height by 21% and 9% in the top 20% of the cloud layer. The impact of droplet evaporation was stronger during *separated* profiles as a greater decrease in humidity (and negative buoyancy) across cloud tops led to greater decreases in median $N_c$ (30%) and LWC (16%) near cloud tops.

Below-cloud $N_a$ was sampled during 61 profiles, and most *contact* profiles (20 out of 28) were within high-$N_a$ (> 350 cm$^{-3}$) boundary layers while most *separated* profiles (22 out of 33) were within low-$N_a$ (< 350 cm$^{-3}$) boundary layers. Although, the differences in below-cloud $N_a$ were statistically insignificant, *contact* profiles within low-$N_a$ boundary layers had up to 34.9 cm$^{-3}$ higher $N_c$ compared to *separated* profiles. This is consistent with weaker impact of droplet evaporation

in the presence of biomass-burning aerosols within 100 m above cloud tops. For *contact* profiles within high-$N_a$ boundary layers, the presence of biomass-burning aerosols led to higher below-cloud $N_a$ (up to 70.5 cm$^{-3}$) and additional droplet nucleation above cloud base along with weaker droplet evaporation. Consequently, the *contact* profiles in high-$N_a$ boundary layers had up to 88.4 cm$^{-3}$ higher $N_c$ compared to *separated* profiles. These results motivate investigations of aerosol-cloud-precipitation interactions over the southeast Atlantic since the changes in $N_c$ and $R_e$ induced by the presence of above-cloud

biomass burning aerosols are likely to impact precipitation rates, liquid water path and cloud fraction, and modulate closed to open cell transitions.

## 1 Introduction

Clouds cover about two-thirds of the Earth's surface (Stubenrauch et al., 2013) and exert a global net cloud radiative effect (CRE) of about – 17.1 W m$^{-2}$ on Earth's energy budget (Loeb et al., 2009). In comparison, the estimated radiative

forcing from 1750 to 2011 due to well-mixed greenhouse gases is +2.83 W m$^{-2}$ (Myhre et al., 2013). The net CRE includes reflection of shortwave solar radiation to space, which cools the Earth, and the absorption (emission) of longwave radiation which warms (cools) the Earth. Marine stratocumulus is a common cloud type that is observed over oceans off western continental coasts where sea-surface temperatures are low and the boundary layer is capped by a strong inversion (Klein and Hartmann, 1993). From 35°S to 35°N, stratocumulus clouds have a shortwave plus longwave top of the atmosphere CRE

between -150 and -200 W m$^{-2}$ with a 10 to 20% contribution to the net CRE (Oreopoulos and Rossow, 2011). General Circulation Models have large uncertainties and inter-model spread in estimates of the net CRE (Boucher et al., 2013). This is partly due to strong underestimation of the subtropical marine stratocumulus cloud cover and the associated CRE (Wang and Su, 2013).

The radiative impact of stratocumulus depends on many factors, including the horizontal and vertical distribution of

cloud droplets, their size distribution and their number concentration. Stratocumulus properties depend on the number, size, composition and vertical distribution of aerosols, and meteorological parameters such as boundary layer height, air mass history and cloud top instability, all of which can modulate the aerosol loading and influence aerosol-cloud interactions. Increases in aerosols acting as cloud condensation nuclei can increase cloud droplet concentration ($N_c$) and decrease effective radius ($R_e$), which increases the cloud optical thickness and shortwave reflectance under conditions of constant

liquid water content (LWC) (Twomey, 1974, 1977). Cloud adjustments in response to this aerosol indirect effect can modulate LWC. For example, precipitation suppression in clouds with smaller droplets increases LWC and cloud lifetime, which increases the CRE (Albrecht, 1989). The indirect effect and rapid adjustments in clouds contribute to the effective radiative forcing due to aerosol-cloud interactions (Boucher et al., 2013). Estimates of the effective radiative forcing (-1.2 to 0.0 W m$^{-2}$) have uncertainties that contribute to the total aerosol radiative forcing, which is "the dominant contributor to

overall net Industrial Era forcing uncertainty" (Myhre et al., 2013).

The impact of the indirect effect can depend on above-cloud thermodynamic parameters such as humidity, buoyancy and inversion strength. Depending on the free tropospheric humidity, dry air entrainment can decrease the LWC in clouds with higher $N_c$ due to the indirect effect (Ackerman et al., 2004; Coakley and Walsh, 2002). Enhanced dry air entrainment can weaken the increase in cloud optical thickness associated with smaller droplets (Small et al., 2009; Rosenfeld et al., 2014). A weak inversion can lead to increased cloud-top entrainment and initiate a stratocumulus-to-cumulus transition by deepening and decoupling the boundary layer, and cutting off the surface moisture source (Wood, 2012). Evaporative cooling from mixing cloudy air with the warm and dry free tropospheric air entraining into clouds leads to cloud-top instability, which is the dominant source of turbulence in stratocumulus (Mellado, 2017).

One of the largest stratocumulus cloud decks on Earth exists off the coast of Namibia over the southeast Atlantic Ocean with a cloud fraction of over 60% between July and October (Devasthale and Thomas, 2011; Zuidema et al., 2016). Biomass-burning aerosols (BBA) that originate from fires in southern Africa (van der Werf et al., 2010) are transported over the stratocumulus by the southern branch of the African Easterly Jet and overlay the clouds (Adebiyi and Zuidema, 2016). The aerosol layer over time descends and mixes with clouds, affecting cloud microphysical properties and their satellite retrievals (Haywood et al., 2004; Costantino and Breon, 2010). Rajapakshe et al. (2017) found the aerosol layer was located within 360 m above the cloud layer for about 60% of the Cloud-Aerosol Transport System (CATS) lidar night-time scenes over the southeast Atlantic. Observations from the NASA ObseRvations of Aerosols above CLouds and their intEractionS (ORACLES) field campaign found the vertical gap between the aerosol layer and cloud tops changed with longitude, having a maximum separation near 7°E, and had a wide range of values (0 to 2,000 m) with near-zero gap for 48% of the scenes (LeBlanc et al., 2020). The southeast Atlantic thus serves as a natural laboratory to examine the effects of varying vertical profiles of above-cloud aerosols on cloud microphysics due to instances of both separation and contact between the BBA layer and the stratocumulus.

BBA over the southeast Atlantic have 500 nm single-scattering albedo ranging between 0.83 and 0.89 (Pistone et al., 2019), which indicates a significant absorbing component to the BBA layer. The warming associated with shortwave absorption by BBA over the southeast Atlantic can be amplified by the evaporation of cloud droplets, the semi-direct effect (Hansen et al., 1997; Ackerman et al., 2000). Aerosols above a reflective cloud layer absorb more solar radiation than aerosols below or within cloud, which affects cloud formation (Haywood and Shine, 1997) and the region's aerosol direct radiative effect (Keil and Haywood, 2003; Cochrane et al., 2019). Shortwave absorption by above-cloud aerosols can increase the buoyancy above cloud tops, inhibit cloud-top entrainment and increase liquid water path (Wilcox, 2010). Large-eddy simulations indicate that the location of the aerosol layer impacts both the magnitude and sign of the semi-direct forcing (Johnson et al., 2004; McFarquhar and Wang, 2006). For example, aerosols above the boundary layer lead to a stronger inversion and decrease entrainment. Additionally, aerosols within the boundary layer cause cloud evaporation and boundary layer decoupling.

The treatment of aerosol effects results in inter-model differences in climate simulations, along with biases in satellite retrievals of clouds and aerosols (Haywood et al, 2004; Brioude et al., 2009; Chand et al., 2009; Coddington et al., 2010;

Painemal and Zuidema, 2011). Many large-scale models do not adequately consider cloud microphysical responses to the vertical separation of aerosols when evaluating aerosol-cloud interactions (Hill et al., 2008). The ORACLES field campaign provides a unique dataset of in-situ observations of cloud and aerosol properties over the southeast Atlantic (Redemann et al., 2021). The impact of above-cloud BBA on stratocumulus properties is quantified by comparing in-situ cloud measurements from instances with layer separation to instances of contact between the aerosol layer and the clouds.

The remainder of the paper is organized as follows. The instrumentation used in the analysis is described in Section 2 along with the procedures for processing the data. A case study of the 6 September 2016 research flight is presented in Section 3. The meteorological and aerosol conditions present are examined and profiles of $N_c$, $R_e$, and LWC are compared for four sawtooth maneuvers flown at locations where clouds were in contact and separated from above-cloud BBA. In Section 4, measurements from six research flights are analysed to investigate buoyancy associated with cloud-top evaporative cooling and profiles of $N_c$, $R_e$, and LWC are compared for boundary layers with similar and varying aerosol loading. Finally, the conclusions and their impact on the understanding of aerosol-cloud interactions are discussed in Section 5.

## 2 Instrumentation

This study presents in-situ measurements of cloud and aerosol properties acquired during the first Intensive Observation Period (IOP) of ORACLES based at Walvis Bay, Namibia (23°S, 14.6°E). The NASA P-3B aircraft conducted research flights west of Africa over the southeast Atlantic Ocean between 1°W to 15°E and 5°S to 25°S from 27 August to 27 September 2016. The aircraft typically flew 50 m to 7 km above the ocean surface and was equipped with in-situ probes for sampling aerosols, clouds and meteorological conditions (Table 1), among other instrumentation. The Passive Cavity Aerosol Spectrometer Probe (PCASP) measured aerosol from approximately 0.1 µm to 3.0 µm using three voltage amplifiers; high, middle and low gain stages (Cai et al., 2013). Laboratory sampling of ammonium sulphate particles conducted after the IOP with the PCASP and a Scanning Mobility Particle Size Spectrometer (SMPS) adjusted the PCASP concentration within each amplification stage to match the measured SMPS concentration. Thereby, a low bias within the middle and high gain stages was corrected to calculate the total aerosol concentration.

A high-resolution time-of-flight aerosol mass spectrometer (HR-ToF-AMS, or AMS) is used to derive the aerosol mass ($M_a$) and chemistry, including organic aerosols (OA) (Table 1). A time- and composition-dependent collection efficiency (CE) was applied to AMS data. The molar ratio of ammonium to sulphate ($NH_4/(2xSO_4)$) was calculated to assess the acidity of liquid aerosol which are collected more efficiently compared to neutralized aerosol. Thus, CE was determined as the maximum between 0.5 and ($1- NH_4/(2xSO_4)$), with a value of 0.5 serving as the lower limit, consistent with estimates from most previous field campaigns (Middlebrook et al., 2012). A Single Particle Soot Photometer (SP2) measured refractory Black Carbon (rBC) concentration and a $CO/CO_2/H_2O$ gas analyzer measured Carbon Monoxide (CO) concentration. The

Spectrometers for Sky-Scanning, Sun-Tracking Atmospheric Research (4STAR) was used to measure column aerosol optical depth (AOD) and retrieve trace gas concentrations above the aircraft (Dunagan et al., 2013; LeBlanc et al., 2020).

The suite of in-situ cloud probes included the Cloud and Aerosol Spectrometer (CAS) on the Cloud, Aerosol and Precipitation Spectrometer (CAPS), Cloud Droplet Probe (CDP), Phase Doppler Interferometer (PDI), 2-Dimensional Stereo Probe (2D-S), Cloud Imaging Probe (CIP) on the CAPS, High Volume Precipitation Sampler (HVPS-3) and the CAPS and King hot-wires. These instruments sampled the droplet number distribution function (n(D)) for droplets with diameters ranging from 0.5 to 19200 µm. The CAPS and King hot-wires measured the bulk LWC. Baumgardner et al. (2017) discuss the general operating characteristics and measurement uncertainties of the in-situ cloud probes and McFarquhar et al. (2017) summarize data processing algorithms. Therefore, only aspects of instrument performance unique to ORACLES 2016 are summarized herein. The in-situ probes used here (CAS, 2D-S, HVPS-3, and PCASP) were calibrated by the manufacturers prior to and shortly after the deployment. During the deployment, performance checks according to the instrument manuals were completed to determine any change in instrument performance. This included monitoring the CAS and 2D-S voltages and temperatures during flights and passing calibration particles through the CAS sample volume to determine any change in the relationship between particle size and peak signal voltage.

CDP data were unusable for the entire 2016 IOP due to an optical misalignment issue. Data from the components of CAPS (CAS, CIP and CAPS hot-wire) were not available before 6 September 2016 because of improper seating of the analog to digital interface board, which resulted in no measurements of droplets less than 50 µm in diameter prior to this flight. The optical lenses were cleaned with isopropyl before each flight, which was especially important during ORACLES since the aircraft frequently flew through aerosol layers that deposited soot on optical lenses of the cloud probes. Stuck bits (photodiodes continuously occluded due to soot deposition) on the optical array probes (2D-S and HVPS-3) were masked during each flight to reduce the presence of artifacts in particle images. The 2D-S vertical channel consistently had photodiode voltages below 1.0 volts due to soot deposition on the inside of the receive-side mirror. Therefore, only data from the horizontal channel are used.

The aircraft's true air speed (TAS) was about 15% higher than the TAS measured by a Pitot tube alongside the CIP. Previous work has shown uncertainties with using the Pitot tube TAS to represent airflow near the probes (Lance et al., 2010; Johnson et al., 2012). Therefore, CAPS, 2D-S, and HVPS-3 probes used the aircraft's TAS, in the absence of reliable TAS measured at these probes' locations. CAPS and PCASP data were processed using the Airborne Data Processing and Analysis processing package (Delene, 2011). 2D-S and HVPS-3 data were processed using the University of Illinois/Oklahoma Optical Probe Processing Software (McFarquhar et al., 2018). Droplets measured by the 2D-S and HVPS-3 having aspect ratios greater than 4 or area ratios less than 0.5 were rejected as artifacts because this study focuses on warm clouds with liquid drops sampled above 0°C. Droplets with inter-arrival times less than 6 µs, indicative of intermittently stuck diodes or drizzle breakup, were removed (Field et al., 2006). Out-of-focus hollow particles were reconstructed following Korolev (2007).

The droplet size distributions from the CAS and 2D-S were merged at 50 µm in diameter to create a combined 1 Hz size distribution, which was used to calculate $N_c$, $R_e$ and LWC. While the HVPS-3 sampled droplets larger than 1280 µm in diameter, only three such 1-s samples, with $N < 0.005$ L$^{-1}$, were sampled during the cloud profiles from the IOP. A threshold of $N_c > 10$ cm$^{-3}$ and bulk LWC $> 0.05$ g m$^{-3}$ for 1 Hz measurements was used to define cloud samples (c.f. Lance et al., 2010; Bretherton et al., 2010). The cloud threshold eliminated the inclusion of optically thinner clouds that a lower LWC threshold of 0.01 g m$^{-3}$ would have included (e.g., Heymsfield and McFarquhar, 2001). Water vapor mixing ratio (q) was determined using a chilled-mirror hygrometer as well as the Los Gatos Research CO/CO$_2$/H$_2$O gas analyzer. The hygrometer suffered from cold soaking during descents from higher elevation and measured lower q near cloud tops during descents compared to ascents into cloud. Measurements of q from the gas analyzer had to be masked for near and in-cloud samples during both ascents and descents due to residual water in the inlet. Therefore, only hygrometer data collected during ascents are used for the analyses involving q.

## 3 Observations on 6 September 2016

### 3.1 Flight track and meteorological conditions

ORACLES research flight tracks included in situ cloud sampling during individual ascents or descents through cloud or during a series of ascents and descents through cloud along a constant heading (sawtooth maneuvers). A case study of the fifth P-3 Research Flight (PRF5) flown on 6 September 2016 was used to examine aerosol and cloud properties sampled under conditions of both contact and separation between the aerosol layer and cloud tops. PRF5 was selected because it had the highest cloud profiling time among the six PRFs with at least eight cloud profiles (Table 2). Four sawtooth maneuvers (S1-S4) were flown during PRF5 (Figure 1) along with four individual cloud profiles (P1-P4). Each sawtooth maneuver consisted of 4-6 individual profiles (Table 2) which were numbered sequentially (e.g., S1-1, S1-2, etc.). South-southeasterly winds (5-8 m s$^{-1}$) were observed at the surface and at 925 mb (Figure 2a, b). This wind field was associated with a surface low-pressure system east of the study region centered around 17°S, 13°E that resulted in advection of low clouds toward the northwest. Open- and closed-cells of marine stratocumulus persisted along with pockets of open cells (POCs) (Figure 1). S1, S2 and S3 were flown along 9°E in closed cells of marine stratocumulus. S4 was flown closer to the coast in a shallow boundary layer with thin closed-cell stratocumulus (Figure 1) later in the day compared to S1-S3 (Figure 3). Ambient temperature sampled by the aircraft sensor was 3 to 6 °C higher during S2 and S3 compared to S1 because the 500 mb geopotential height and relative humidity (RH) were higher toward the north (Figure 2b). Cloud top height ($Z_T$) is identified as the highest altitude satisfying the criteria used to define cloud ($N_c > 10$ cm$^{-3}$ and bulk LWC $> 0.05$ g m$^{-3}$). S1, S2 and S3 had higher $Z_T$ compared to S4 (Figure 3) due to the advection of cold, dry continental air from the southeast and low RH ($<$ 70%) where S4 was flown which resulted in cloud thinning and a shallower boundary layer (Figure 2b, c).

The aircraft intermittently entered and exited cumulus clouds below the stratocumulus layer during 33 of the 71 cloud profiles flown during the IOP (Table 2) which resulted in fluctuating values of $N_c$ and $R_e$, with bulk LWC $< 0.05$ g m$^{-3}$. For

example, during S1-3, $N_c$ varied between 10 to 240 cm$^{-3}$ and $R_e$ varied between 3 to 12 µm up to 130 m below where the stratocumulus base was identified with bulk LWC > 0.05 g m$^{-3}$. Images from a forward-facing camera on the aircraft contrast a boundary layer with multiple cloud layers (Figure 4a; image taken at 08:53 UTC) during S1-3 and a shallow, well-mixed boundary layer capped by stratocumulus (Figure 4b; image taken at 13:16 UTC) during S4-1. It is likely the stratocumulus layer was decoupled from the surface where S1-3 was flown because the boundary layer was deepened by the entrainment of free tropospheric air. Subsequently, the sub-cloud layer was well-mixed with the surface and topped by shallow cumulus similar to observations by Wood (2012). The cloud base height ($Z_B$) for the 33 profiles was determined as the lowest altitude with $N_c$ > 10 cm$^{-3}$ and bulk LWC > 0.05 g m$^{-3}$ above which a continuous cloud layer was sampled. S4 had lower $Z_B$ (195-249 m) compared to S1 (676-691 m), S2 (534-598 m) and S3 (501-775 m) (Figure 3).

### 3.2 Above- and below-cloud aerosol composition

For each sawtooth maneuver, the above- and below-cloud air mass source region was identified using five-day back-trajectories computed using the NOAA Hybrid Single Particle Lagrangian Integrated Trajectory model (Stein et al., 2015) applied to the National Center for Environmental Prediction Global Data Assimilation System model (Figure 5). The concentrations listed in Table 3 indicate measurements up to 100 m above and below the clouds averaged across the cloud profiles for each sawtooth maneuver. The variability in above-cloud $M_a$ and $N_a$ for S1-S4 was driven by the above-cloud air mass source region. The above-cloud air mass sampled near S1 and S4 originated from the boundary layer from the southeast and the above-cloud air mass sampled near S2 and S3 descended from higher altitudes over the African continent (Figure 5b, c). The above-cloud OA $M_a$ and $N_a$ for S2 and S3 were over 5 times higher than the corresponding values for S1 and S4 (Table 3). The below-cloud air mass sampled during S1-S4 was advected from the boundary layer from the southeast (Figure 5a, c). During S1 and S4, the above- and below-cloud rBC and CO concentrations were similar (Table 3) since the above-cloud air mass also originated from the south east (Figure 5b, c). During S2 and S3, the continental above-cloud air mass had much higher rBC and CO (over 500 cm$^{-3}$ and 190 ppb) compared to the below-cloud air mass from the south east (below 150 cm$^{-3}$ and 120 ppb). Since OA, rBC and CO are indicators of combustion, this suggests the continental above-cloud air mass had greater exposure to biomass-burning products compared to the air masses from the south east. S2 and S3 also had higher below-cloud rBC and CO compared to S1 and S4 (Table 3) which suggests the BBA with high $N_a$ within 100 m above clouds could be mixing into the cloud layer and polluting the boundary layer. This is also likely to be associated with the history of entrainment mixing of polluted free tropospheric air into the boundary layer prior to these observations (Diamond et al., 2018).

### 3.3 Cloud profile classification

Every sawtooth maneuver was preceded by a 5 to 10-minute constant-altitude flight leg about 100 m above the cloud layer to retrieve the above-cloud aerosol optical depth (AOD) using 4STAR. Average above-cloud AOD at 550 nm within 50 km of the sampling locations for S1-S4 ranged between 0.33 and 0.49, indicating a BBA layer was located at some

altitude above the clouds sampled during S1-S4. During S1, above-cloud $N_a < 500$ cm$^{-3}$ was sampled up to 200 m above cloud tops (Figure 3) which indicates the BBA layer was separated from cloud tops. During S4, the level of above-cloud $N_a$

$> 500$ cm$^{-3}$ was identified over 200 m above cloud tops indicating a similar separation. Therefore, cloud profiles flown during S1 and S4 were classified as *separated* profiles. During S2 and S3, the level of above-cloud $N_a > 500$ cm$^{-3}$ was located within 100 m above cloud tops and the BBA layer was likely in contact with the cloud tops. Therefore, cloud profiles flown during S2 and S3 were classified as *contact* profiles. In a previous study, a significantly higher threshold (PCASP $N_a =$ 1000 cm$^{-3}$) was used to identify the BBA layer above stratocumulus clouds off the coast of California (Mardi et al., 2018).

The sensitivity of the threshold chosen in this study is examined in Appendix-A and using a threshold of 1000 cm$^{-3}$ would have no significant impact on the results presented in this study.

### 3.4 Vertical profiles of $N_c$, $R_e$, and LWC

Since $Z_B$ and cloud thickness (H) varied between profiles, $N_c$, $R_e$, and LWC were examined as a function of normalized height above cloud base ($Z_N$), where $Z_N = (Z - Z_B)/(Z_T - Z_B)$ and varied from 0 (cloud base) to 1 (cloud top). Measurements

from the four sawtooth maneuvers were compared following McFarquhar et al. (2007), and divided into 10 $Z_N$ bins where each bin represented 10% of the cloud layer (Figure 6). For example, the bin with $0 < Z_N < 0.1$ (represented by the midpoint, $Z_N = 0.05$) included data collected over the bottom 10% of the cloud layer. For *separated* profiles, droplet nucleation occurred near cloud base with the median $N_c$ increasing up to $Z_N = 0.25$ (S1: 132 to 179 cm$^{-3}$, S4: 23 to 85 cm$^{-3}$). The impact of droplet nucleation decreased above cloud base ($Z_N = 0.25$ to 0.75) and median $N_c$ increased by up to 30 cm$^{-3}$ for S1 and

decreased by up to 15 cm$^{-3}$ for S4 (Figure 6a). Condensational growth occurred over these levels as the median $R_e$ increased with $Z_N$ (Figure 6b). The median $N_c$ decreased near cloud top ($Z_N = 0.75$ to 0.95) due to droplet evaporation resulting from cloud-top entrainment mixing between cloudy and non-cloudy air. *Contact* profiles (S2 and S3) had higher median $N_c$ at cloud base compared to *separated* profiles which decreased with height up to $Z_N = 0.25$ (S2: 190 to 169 cm$^{-3}$, S3: 180 to 131 cm$^{-3}$). The median $N_c$ for S2 and S3 increased by up to 43 cm$^{-3}$ over $Z_N = 0.25$ to 0.75 and decreased near cloud top due to

droplet evaporation. S4 had the lowest $N_c$ at cloud base because the below-cloud $M_a$ and $N_a$ for S4 were over a factor of 3 lower than the corresponding values for S1-S3 (Table 3).

Consistent with condensational growth and collision-coalescence, median $R_e$ increased with $Z_N$ from cloud base to top, from 6.0 µm to 6.7 µm, 4.6 to 6.9 µm, 4.9 to 8.3 µm and 8.7 to 9.9 µm for S1-S4, respectively (Figure 6b). S1 and S4 had higher median $R_e$ at cloud base due to higher drizzle (droplets with diameters larger than 50 µm) concentrations (41 and 31

L$^{-1}$) compared to S2 and S3 (14 and 18 L$^{-1}$). For S4, drizzle concentration decreased from $Z_N = 0.05$ to 0.25 which led to the decrease in median $R_e$ over these heights. The median LWC increased with height up to at least $Z_N = 0.75$ and decreased near cloud tops due to droplet evaporation (Figure 6c). The LWC for each sawtooth maneuver was lower than the adiabatic LWC (aLWC) due to cloud-top entrainment mixing and the ratio of LWC to aLWC was used to quantify the degree of mixing. Lower LWC/aLWC (averaged over the cloud layer) for S2 and S3 (0.37 and 0.41) compared to S1 and S4 (0.51 and

0.55) indicated that *contact* profiles had greater mixing between cloudy and non-cloudy air in the cloud layer, on average.

The boundary layer was capped by an inversion with warmer, drier air above the clouds. During S1-S4, the temperature increased above cloud top by 10.3, 9.3, 8.9 and 1.5°C, and the total water mixing ratio decreased by 6.2, 5.4, 2.3 and 0.4 g kg$^{-1}$, respectively (Figure 7). The decreases in $N_c$ and LWC near stratocumulus tops have been attributed to cloud-top entrainment of the overlying warm and sub-saturated air (Wood, 2012). Droplet evaporation due to the entrainment mixing resulted in decreases of 14%, 28%, 12% and 26% in the median $N_c$ near cloud tops during S1-S4, respectively.

### 3.5 Evidence of the aerosol indirect effect

$N_c$ and $R_e$ were compared between sawtooth maneuvers and the differences reported hereafter refer to 95% confidence intervals for the difference in the variable means (based on a two-sample t-test, $p < 0.02$). Between the *contact* profiles, S2 had significantly higher $N_c$ (differences of 37 to 56 cm$^{-3}$) compared to S3. This was despite having statistically insignificant differences in below-cloud $N_a$, a greater fractional decrease in median $N_c$ near cloud top compared to S3, and greater entrainment mixing (lower LWC/aLWC). S2 had significantly higher above-cloud $N_a$ compared to S3 and the mixing of above-cloud air with high $N_a$ likely resulted in droplet nucleation above cloud base, where the median $N_c$ for S2 increased from 169 to 220 cm$^{-3}$ over $Z_N = 0.25$ to 0.75. Between the *separated* profiles, S1 had significantly higher $N_c$ (differences of 108 to 126 cm$^{-3}$) which could be attributed to significantly higher above-cloud $N_a$ and greater entrainment mixing during S1 compared to S4. However, these differences could also be due to the meteorological differences at their sampling locations (lower boundary layer height, RH and 500 mb geopotential height, $\Delta T$ and $\Delta q_T$ for S4) or the significantly higher below-cloud $N_a$ for S1 compared to S4.

*Contact* profiles had significantly higher $N_c$ (differences of 45 to 61 cm$^{-3}$) and lower $R_e$ (differences of 1.4 to 2.0 µm) compared to *separated* profiles. *Contact* profiles also had significantly higher above-cloud $N_a$ and greater entrainment mixing in the cloud layer (lower LWC/aLWC). These microphysical changes would also impact cloud reflectance (Twomey, 1991) as seen by the significantly higher cloud optical thickness ($\tau$) of *contact* profiles compared to *separated* profiles (differences of 2.5 to 8.2). The increase in $\tau$ and the cloud reflectance provides observational evidence of the aerosol indirect effect over the southeast Atlantic due to contact between above-cloud BBA and the stratocumulus clouds.

However, *contact* profiles also had significantly higher below-cloud $N_a$ (differences of 145 to 190 cm$^{-3}$) which contribute to the higher $N_c$ relative to *separated* profiles. Therefore, a statistical analysis was conducted with a larger number of profiles in an attempt to attribute these differences in $N_c$ and $R_e$ to the vertical distance between the above-cloud BBA layer and cloud tops. Building on this case study, 71 cloud profiles flown on six flights between 6 and 25 September 2016 were examined and the impact of above-cloud BBA on the free tropospheric humidity and buoyancy across cloud tops was explored. 61 *contact* and *separated* profiles were further classified as low-$N_a$ or high-$N_a$ profiles based on the below-cloud $N_a$. This was done to quantify the differences in $N_c$ and $R_e$ between *contact* and *separated* profiles within boundary layers with similar below-cloud $N_a$.

## 4 Statistical Analysis

### 4.1 Meteorological conditions and above-cloud aerosols

Six flights (including PRF5) are included in the statistical analysis. On 10, 12 and 25 September, the P-3 took off from Walvis Bay, Namibia (23˚S, 14.6˚E) and flew north-west from 23˚S, 13.5˚E toward 10˚S, 0˚E, returning along the same track (Figure 8). Different tracks were followed on 6, 14 and 20 September which included meridional legs along 9˚E, 7.5°E and 9°E, and 9°E and 10.5°E, respectively. Meteorological conditions on 10, 12 and 14 September were similar to the conditions described for the case study. South-southeasterly surface winds were associated with a surface low-pressure system over Africa. The surface wind speeds varied between 5 to 10 m s$^{-1}$ depending on the pressure gradient between the continental low and a surface high toward the southwest. A region of 925 mb RH < 60% persisted along the coast due to dry air advection from Africa. A different meteorological setup on 20 September had westerly surface winds and easterly winds at 925 mb. The aerosol plume was sampled immediately above the boundary layer (600 m) as warm surface air was overlaid by drier, polluted air from the continent. The continental surface low was located farther south on 25 September compared to other flight days with the region of low 925 mb RH to the south of the flight track. The study region had RH >60% with south-southeasterly surface winds and southerly 925 mb winds. The BBA layer with above-cloud $N_a$ > 500 cm$^{-3}$ was sampled during each flight with variability in its vertical location (Table 4). Only *separated p*rofiles were flown on 10 and 14 September (Table 2) when the BBA layer and cloud tops were separated by over 600 and 1500 m, respectively (Table 4). On 12 September, Profile 1 (P1) had $N_a$ > 500 cm$^{-3}$ within 75 above cloud tops and was classified as a *contact* profile while P2 and S1 were classified as *separated* profiles. On 20 September, each profile had above-cloud AOD > 0.4 and was classified as a *contact* profile. On 25 September, the profiles had above-cloud AOD > 0.27 and each profile (except from a sawtooth near 11°S, 1°E) was classified as a *contact* profile.

### 4.2 $N_c$, $R_e$, and LWC for *contact* and *separated* profiles

Since clouds sampled on different flight days had variable $Z_B$ and $Z_T$ (Figure 9), vertical profiles of $N_c$, $R_e$ and LWC from the *contact* and *separated* profiles were compared as a function of $Z_N$. The frequency distributions of $N_c$, $R_e$ and LWC as a function of $Z_N$ are examined in Fig. 10 using violin plots (Hintze and Nelson, 1998; Wang et al., 2020) where the width of the shaded area represents the proportion of data there. The average $N_c$ for *contact* profiles was significantly higher than the average $N_c$ for *separated* profiles (differences of 60 to 68 cm$^{-3}$). During *separated* profiles, the median $N_c$ had little variability up to $Z_N$ = 0.75 (114 to 122 cm$^{-3}$) and decreased thereafter with $Z_N$ to 73 cm$^{-3}$ due to droplet evaporation (Figure 10a). During *contact* profiles, the median $N_c$ decreased slightly up to $Z_N$ = 0.25 (183 to 174 cm$^{-3}$), increased to 214 cm$^{-3}$ at $Z_N$ = 0.75, and decreased near cloud top to 157 cm$^{-3}$ due to droplet evaporation. *Contact* profiles had significantly lower $R_e$ than the *separated* profiles (differences of 1.1 to 1.3 μm) and the median $R_e$ increased with $Z_N$ from 4.9 to 7.0 μm for *contact* and from 6.6 to 8.6 μm for *separated* profiles (Figure 10b). The differences in $R_e$ were likely due to the significantly lower drizzle concentrations for *contact* profiles (differences of 5 to 20 L$^{-1}$).

The average LWC for *contact* and *separated* profiles were within 0.01 g m$^{-3}$, and the median LWC increased with $Z_N$ to

0.23 g m$^{-3}$ at $Z_N$ = 0.85 for *contact* and 0.21 g m$^{-3}$ at $Z_N$ = 0.75 for *separated* profiles (Figure 10c). *Contact* profiles had lower LWC/aLWC in the cloud layer (0.45) compared to *separated* profiles (0.57) which suggests there was greater entrainment mixing during *contact* profiles, on average. However, droplet evaporation near cloud top had a stronger impact on *separated* profiles as the median LWC decreased to 0.16 g m$^{-3}$ for *separated* and 0.20 g m$^{-3}$ for *contact* profiles (Figure 10c). *Separated* profiles had a greater decrease in LWC/aLWC near cloud top (0.41 to 0.26) compared to *contact* profiles

(0.38 to 0.30) and greater fractional decreases in median $N_c$ and LWC (40% and 16%) compared to *contact* profiles (25% and 9%). The stronger impact of droplet evaporation during *separated* profiles contributed to the differences between $N_c$ for *contact* and *separated* profiles.

### 4.3 Cloud-top Evaporative Cooling

Buoyancy and humidity across cloud tops were determined to explore the cloud-top entrainment mechanisms resulting

in the differential impact of droplet evaporation for these profiles. Cloud-top instability is the dominant source of turbulence in stratocumulus with evaporative cooling being a key driver of instability (Mellado, 2017). Recent studies have shown there is strong correlation between above-cloud AOD and water vapor within air masses originating from the African continent (Deaconu et al., 2019; Pistone et al., 2021). Longwave cooling by water vapor within the BBA layer leads to decreased cloud-top cooling and cloud-top dynamics are influenced by distinct radiative contributions from water vapor and absorbing

aerosols. Evaporative cooling in a mixture of dry and cloudy air near cloud top generates negatively buoyant air mixtures which further enhances mixing and leads to an entrainment feedback called Cloud Top Entrainment Instability or CTEI (Kuo and Schubert, 1988). Under such conditions, negative buoyancy leads to an unstable feedback, unlike the conventional association of negative buoyancy with atmospheric stability. The critical condition for cloud-top stability is given by Kuo and Schubert (1988) as

$$\Delta\theta_e > k \left(\frac{L_v}{c_p}\right)\Delta q_T \quad , \tag{1}$$

where k is the CTEI parameter, $\theta_e$ is the equivalent potential temperature, $L_v$ is the latent heat of vaporization, and $c_p$ is the specific heat capacity of air at constant pressure. The $\Delta$ operator represents gradients across the cloud-top, defined here as the difference between $\theta_e$ (or $q_T$) measured 100 m above cloud top and the vertical average of $\theta_e$ (or $q_T$) over the top 100 m of the profile. Following Eq. (13) from Kuo and Schubert (1988), k > 0.23 indicates negative buoyancy across cloud tops.

Water vapor mixing ratio (q) measured by the chilled-mirror hygrometer was used to calculate $\theta_e$ and $q_T$. Since lower $\Delta q_T$ was sampled during descents into cloud due to condensation on the hygrometer, k-values for descents were determined to be measurement artifacts and not usable here.

All *separated* profiles (except PRF5 S1-3 and S4-1, 3, 5) laid within the region of cloud-top instability (k > 0.23) on a $\Delta\theta_e$ - $\Delta q_T$ plane (Figure 11) and showed negative buoyancy across cloud tops. During PRF5 S1-3, low $\Delta\theta_e$ was sampled due

to higher above-cloud humidity associated with the presence of $N_a$ > 100 cm$^{-3}$ within 50 m above cloud tops. During PRF5

S4, a weak cloud-top inversion led to positive $\Delta\theta_e$ and $\Delta q_T < -2$ g kg$^{-1}$ (Fig. 7). For the remaining *separated* profiles, negative buoyancy across cloud tops led to forced descent of dry, free tropospheric air into the clouds. Since the free tropospheric air was warmer and drier than the cloudy air, droplet evaporation led to the decreases in median $N_c$ and LWC near cloud top. The positive evaporative cooling feedback and greater $\Delta q_T$ compared to *contact* profiles (Figure 11) explain the stronger impact of droplet evaporation on median $N_c$ and LWC for *separated* profiles. While evaporative cooling triggered the CTEI feedback, the clouds persisted consistent with cloud-top radiative cooling or surface evaporation leading to boundary layer moistening (Lock, 2009; Mellado, 2017).

All *contact* profiles (except PRF13 S1-3) laid within the region of cloud-top stability and showed positive buoyancy across cloud tops. Entrainment mixing for these profiles likely occurred when the clouds penetrated the inversion. This is consistent with significantly higher average H (267 m) for *contact* profiles compared to *separated* profiles (213 m). Braun et al. (2018) found a negative correlation between H and adiabaticity (ratio of the measured and the adiabatic liquid water path) which is consistent with *contact* profiles having lower LWC/aLWC and higher H compared to *separated* profiles. In the presence of above-cloud BBA, the above-cloud air was more humid, and the above-cloud $N_a$ were significantly higher compared to *separated* profiles (differences of 768 to 831 cm$^{-3}$). *Contact* profiles had greater entrainment mixing compared to *separated* profiles and the median $N_c$ increased with height over $Z_N = 0.25$ to 0.75. It is likely the entrainment of BBA into clouds resulted in additional droplet nucleation over these $Z_N$ levels. Therefore, weaker droplet evaporation near cloud top and additional droplet nucleation above cloud base in the presence of above-cloud BBA likely contributed to the differences between $N_c$ for *contact* and *separated* profiles.

### 4.4 $N_c$, $R_e$ and LWC in boundary layers with similar $N_a$

*Contact* profiles had significantly higher below-cloud $N_a$ (differences of 93 to 115 cm$^{-3}$) and below-cloud CO (differences of 13 to 16 ppb) in addition to higher above-cloud $N_a$ (differences of 768 to 831 cm$^{-3}$) compared to *separated* profiles. Enhanced aerosol loading within the boundary layer is consistent with BBA immediately above cloud tops entraining into the cloud layer and polluting the boundary layer. This is consistent with higher above-cloud CO (240 ppb) sampled for *contact* profiles with below-cloud CO > 100 ppb compared to above-cloud CO (104 ppb) for profiles with below-cloud CO < 100 ppb. The correlations between above- and below-cloud aerosols could be partly due to the history of entrainment mixing between free tropospheric and boundary layer air masses (Diamond et al., 2018). To investigate the contribution of below-cloud $N_a$ relative to the impact of above-cloud BBA on cloud properties, 28 *contact* and 33 *separated* profiles were classified into four new regimes defined as follows: . Contact-high $N_a$ (C-H), Separated-high $N_a$ (S-H), Contact-low $N_a$ (C-L), and Separated-low $N_a$ (S-L), where high- and low-$N_a$ boundary layers were separated using a threshold concentration of 350 cm$^{-3}$. Cloud microphysical properties and above/below-cloud $N_a$ were compared between 20 C-H and 11 S-H profiles and between 8 C-L and 22 S-L profiles (Table 5) to compare *contact* and *separated* profiles with minor differences in below-cloud $N_a$.

Within low-$N_a$ boundary layers, C-L and S-L profiles had insignificant differences in below-cloud $N_a$ despite significantly higher above-cloud $N_a$ for C-L profiles (differences of 592 to 669 cm$^{-3}$), higher $N_c$ (differences of 22.8 to 34.9 cm$^{-3}$) and lower $R_e$ (differences of 0.5 to 1.0 µm) compared to S-L profiles. Within high-$N_a$ boundary layers, C-H profiles had significantly higher below-cloud $N_a$ compared to S-H profiles (differences of 39.1 to 70.5 cm$^{-3}$), but the differences were much smaller than those in the above-cloud $N_a$ (differences of 738 to 884 cm$^{-3}$). Further, the C-H profiles had significantly higher $N_c$ (differences of 75.5 to 88.5 cm$^{-3}$) and lower $R_e$ (differences of 1.1 to 1.3 µm) than the S-H profiles. Previous studies have argued the changes in $N_c$ due to the impact of BBA are more strongly correlated with below-cloud $N_a$ compared to above-cloud $N_a$ (Diamond et al., 2018; Mardi et al., 2019). However, these results suggest that although the differences in $N_c$ were lower than the differences in above-cloud $N_a$, significant changes in $N_c$ and $R_e$ were associated with contact with above-cloud BBA, and these changes were independent of the below-cloud aerosol loading.

Vertical profiles of $N_c$, $R_e$, and LWC are examined (Fig. 12) to further investigate the microphysical changes due to contact with above-cloud BBA. Within low-$N_a$ boundary layers, there were minor deviations in $N_c$ with $Z_N$ up to $Z_N = 0.75$ (Figure 12a). Over the top 20 % of the cloud layer, S-L profiles had a decrease in median $N_c$ (32 cm$^{-3}$) with a smaller change for C-L profiles (8 cm$^{-3}$) over the same levels. There was also a weaker decrease in water vapor mixing ratio across cloud tops for *contact* profiles. Thus, cloud-top entrainment of more humid air likely occurred for the C-L profiles. This is consistent with higher median $R_e$ and LWC over $Z_N = 0.75$ to 0.95 for C-L profiles compared to S-L profiles despite having lower $R_e$ and LWC closer to cloud base (Figure 12b, c). Thus, the microphysical differences between *contact* and *separated* profiles within low-$N_a$ boundary layers (where most *separated* profiles were sampled) are consistent with the processes of cloud-top entrainment and droplet evaporation.

The differences between below-cloud $N_a$ for C-H and S-H profiles (39.1 to 70.5 cm$^{-3}$) were lower than the corresponding differences in $N_c$ (75.5 to 88.4 cm$^{-3}$). C-H profiles had significantly higher $N_c$ and lower $R_e$ compared to S-H profiles throughout the cloud layer (Figure 12a, b). There was a significant increase in median $N_c$ for C-H profiles over $Z_N = 0.25$ to 0.75 which was accompanied by higher median LWC for C-H profiles in the top half of the cloud layer. This is consistent with additional droplet nucleation above cloud base during C-H profiles. Additionally, there was a stronger decrease in $N_c$ near cloud top for S-H profiles ($N_c$ decreased by 66 cm$^{-3}$) compared to C-H profiles ($N_c$ decreased by 29 cm$^{-3}$) likely due to cloud-top entrainment. It is difficult to separate the impact of changes in droplet nucleation on differences in $N_c$ between C-H and S-H profiles from the impact of changes in droplet evaporation due to cloud-top entrainment. Therefore, it is speculated the microphysical changes within high-$N_a$ boundary layers were likely driven by the combination of higher below-cloud $N_a$, potential droplet nucleation above cloud base, and weaker droplet evaporation near cloud tops in the presence of above-cloud BBA. The sensitivity of these results to using different thresholds to locate BBA (other than 500 cm$^{-3}$), to define "separation" between the aerosol and cloud layers (other than 100 m), and to define a "high-$N_a$ boundary layer" (other than 350 cm$^{-3}$) is discussed in Appendix – A, but does not affect the qualitative findings.

## 5 Discussion

The presence of water vapor and absorbing aerosols within the BBA layer can have distinct impacts on cloud-top cooling and cloud-top dynamics (Deaconu et al., 2019; Herbert et al., 2020; Kuo and Schubert, 1988). In the presence of above-cloud BBA during ORACLES, the above-cloud air was more humid than in its absence, and cloud-top entrainment of free tropospheric air with higher water vapor mixing ratio likely contributed to the microphysical differences between *contact* and *separated* profiles, consistent with previous observations (Ackerman et al., 2004). Further, C-H profiles had significantly lower drizzle concentration compared to S-H profiles (differences of 4 to 21 $L^{-1}$) but C-L and S-L profiles had similar drizzle concentrations (61 $L^{-1}$ and 62 $L^{-1}$). Research is ongoing to examine the changes in cloud and precipitation properties in different aerosol regimes since precipitation suppression could also impact below-cloud $N_a$ through reduced aerosol scavenging by drizzle (Pennypacker et al., 2020).

Within polluted boundary layers, the below-cloud $N_a$ was larger for instances of contact between above-cloud BBA and cloud tops. It is speculated the increase in below-cloud $N_a$ alone would be insufficient to cause the microphysical differences between *contact* and *separated* profiles, and this is particularly true for polluted boundary layers. The $N_c$ also depends on other factors including updraft strength and aerosol composition and hygroscopicity (Fuchs et al., 2018; Kacarab et al., 2020; Mardi et al., 2019). High-resolution modelling studies with bin-resolved microphysics are needed to examine cloud-top entrainment processes and investigate the relative impact of semidirect and indirect effects of BBA on marine stratocumulus over the southeast Atlantic. Additionally, aerosol-cloud-precipitation interactions must be examined under different aerosol and meteorological regimes to investigate the buffering effects of local meteorology and thermodynamic profiles associated with the absorbing aerosols (Deaconu et al., 2019; Diamond et al., 2018; Fuchs et al., 2018; Herbert et al., 2020; Sakaeda et al., 2011; Stevens and Feingold, 2009).

The changes in $N_c$, $R_e$, and drizzle concentration presented here could lead to aerosol-induced precipitation suppression and impact stratocumulus to cumulus transitions over the southeast Atlantic (Yamaguchi et al., 2015; Zhou et al., 2017). Subsequently, changes in precipitation rate could affect the balance between aerosol scavenging and entrainment and modulate the reversible open-closed cell transitions (Abel et al., 2020; Feingold et al., 2015). These processes would affect the cloud radiative forcing and the direct aerosol radiative forcing which depends on the albedo of the underlying cloud layer (Cochrane et al., 2019). Research is ongoing to quantify precipitation susceptibility as a function of the vertical displacement of above-cloud absorbing aerosols from cloud tops. A larger dataset including additional ORACLES observations from August 2017 and October 2018 will allow evaluation of cloud and precipitation retrievals (Dzambo et al., 2019; Painemal et al., 2020) and investigations of aerosol-cloud-precipitation interactions over a broader range of environmental conditions. Better understanding of these processes will help reduce uncertainties in the estimates of cloud radiative effects due to changes in cloud cover and cloud reflectance (Albrecht, 1989; Twomey, 1974; Twomey, 1991).

**6 Conclusions**

This study provides observational evidence of the aerosol indirect effect on marine stratocumulus cloud properties due to contact between above-cloud biomass burning aerosols and stratocumulus cloud tops over the southeast Atlantic Ocean. Biomass-burning aerosols overlay marine stratocumulus clouds there with variability in the vertical separation (0 to 2000 m) between the aerosol layer and cloud tops. In-situ measurements of cloud and aerosol properties from 6 research flights during the NASA ORACLES field campaign in September 2016 are presented. These observations suggest the presence of biomass-burning aerosols immediately above cloud tops was associated with changes in vertical profiles of $N_c$, $R_e$, and LWC due to cloud-top entrainment and increases in the free tropospheric temperature and humidity. Meteorological conditions and the vertical profiles of $N_c$, $R_e$, LWC and above- and below-cloud $N_a$ are examined for a case study of 6 September 2016. Thinner clouds with lower cloud base and top heights were sampled closer to the coast due to lower relative humidity and boundary layer height compared to clouds sampled along 9°E. For 33 cloud profiles, cloud-top entrainment deepened the boundary layer, decoupled the stratocumulus layer from the surface and resulted in cumulus formation below the stratocumulus. The vertical profiles of cloud ($N_c$, $R_e$ and LWC) and thermodynamic ($q_T$ and T) properties sampled on 6 September 2016 were consistent with observations of stratocumulus-topped boundary layers capped by an inversion with warm, dry free tropospheric air above the clouds (Wood, 2012).

Above-cloud air masses originating from Africa were composed of biomass-burning products (OA, rBC and CO) with higher $N_a$ compared to above-cloud air masses originating from the boundary layer over the southeast Atlantic Ocean. 30 *contact* profiles were flown where the level of $N_a > 500$ cm$^{-3}$ was within 100 m above cloud tops and 41 *separated* profiles were flown where $N_a > 500$ cm$^{-3}$ was sampled at least 100 m above cloud tops. For *contact* profiles, the average $N_c$ in the cloud layer was up to 68 cm$^{-3}$ higher, the average $R_e$ was up to 1.3 µm lower, and the average LWC was within 0.01 g m$^{-3}$ compared to *separated* profiles. During the *contact* profiles, $q_T$ decreased across cloud tops by up to 6 g kg$^{-1}$. With positive buoyancy across cloud tops, mixing between free tropospheric and cloudy air occurred when clouds penetrated the inversion and median $N_c$ and LWC decreased by 25% and 9% near cloud tops due to droplet evaporation. The entrainment mixing of free tropospheric air with $N_a > 500$ cm$^{-3}$ likely resulted in droplet nucleation above cloud base and the median $N_c$ for *contact* profiles increased within the middle of the cloud layer. During *separated* profiles, $q_T$ decreased across cloud tops by up to 9 g kg$^{-1}$. With negative buoyancy across cloud tops, forced descent of drier free tropospheric air into the clouds resulted in a positive feedback of evaporative cooling, and median $N_c$ and LWC decreased by 30% and 16% due to droplet evaporation. The median $N_c$ during *separated* profiles had little variability with height above cloud base before decreasing near cloud top due to droplet evaporation. Therefore, *contact* profiles had higher $N_c$ due to a combination of weaker droplet evaporation near cloud tops and additional droplet nucleation above cloud base in the presence of above-cloud biomass-burning aerosols.

Biomass-burning aerosols located immediately above cloud top mixed into the cloud and polluted the boundary layer. During the case study, sawtooth maneuvers with *contact* profiles had higher below-cloud rBC and CO concentrations (by up to 60 cm$^{-3}$ and 30 ppb) compared to maneuvers with *separated* profiles. Among the 71 profiles across six research flights,

contact profiles had significantly higher below-cloud CO and $N_a$ compared to *separated* profiles due to the contact between biomass-burning aerosols and cloud tops. 28 *contact* and 33 *separated* profiles were further classified as Contact-high $N_a$ (C-H), Contact-low $N_a$ (C-L), Separated-high $N_a$ (S-H) and Separated-low $N_a$ (S-L) to represent *contact* or *separated* profiles within high-$N_a$ (> 350 cm$^{-3}$) or low-$N_a$ (< 350 cm$^{-3}$) boundary layers. C-L profiles had up to 34.9 cm$^{-3}$ higher average $N_c$ and up to 0.9 μm lower average $R_e$ compared to S-L profiles despite statistically insignificant differences between the below-cloud $N_a$. C-H profiles had up to 70.5 cm$^{-3}$ higher below-cloud $N_a$, up to 88.4 cm$^{-3}$ higher $N_c$, and up to 1.6 μm lower $R_e$ compared to S-H profiles. The differences between *contact* and *separated* profiles in low-$N_a$ boundary layers were likely driven by weaker droplet evaporation in the presence of above-cloud biomass-burning aerosols. Within high-$N_a$ boundary layers, the median $N_c$ increased with height in the middle of the cloud layer, potentially due to droplet nucleation above cloud base. The differences between *contact* and *separated* profiles within high-$N_a$ boundary layers were likely driven by a combination of higher below-cloud $N_a$, droplet nucleation above cloud base, and weaker droplet evaporation in the presence of biomass-burning aerosols above cloud tops.

**Appendix - A**

Cloud profiles were classified as *contact* or *separated* according to whether above-cloud $N_a$ greater than 500 cm$^{-3}$ was measured at a level within 100 m above cloud tops. The classification of cloud profiles remained unchanged when $N_a$ = 400 cm$^{-3}$ instead of $N_a$ = 500 cm$^{-3}$ was used to locate the aerosol layer. When the level of $N_a$ = 300 cm$^{-3}$ was used, 3 of the 26 *separated* profiles (PRF5 S1, PRF5 P2 and PRF7 P6) switched to the *contact* regime. The qualitative results were unchanged as *contact* profiles had higher $N_c$ (differences of 63 to 71 cm$^{-3}$) and lower $R_e$ (differences of 1.1 to 1.3 μm) compared to *separated* profiles. When a level of $N_a$ = 600 cm$^{-3}$ was used, 2 of the 15 *contact* profiles (PRF5 P1 and P3) switched to the *separated* regime and *contact* profiles had higher $N_c$ (differences of 59 to 67 cm$^{-3}$) and lower $R_e$ (differences of 1.0 to 1.2 μm). No additional changes were observed upon changing the definition of the BBA layer. Thus, the results obtained were robust as relates to this threshold.

A gap of 100 m was used to define separation between the BBA and the clouds. When this gap was decreased to 50 m, 4 of the 15 *contact* profiles (PRF5 P4, PRF8 P1 and PRF11 S1, P6) switched to the *separated* regime and the *contact* regime had higher $N_c$ (differences of 50 to 59 cm$^{-3}$) and lower $R_e$ (differences of 0.67 to 0.92 μm). There was no change in the profile classification when increasing the gap from 100 m to 200 m. On increasing the gap to 300 m, PRF5 S4 switched to the *contact* regime and contact profiles had higher $N_c$ (differences of 36 to 45 cm$^{-3}$) and lower $R_e$ (differences of 0.4 to 0.6 μm). The same profile switches were observed when the definition of the gap was varied between 50 and 300 m for a threshold of above-cloud $N_a$ = 400 cm$^{-3}$ to locate the BBA layer. Thus, the findings were robust as relates to the choice of these thresholds.

There were no profiles with maximum below-cloud $N_a$ < 100 cm$^{-3}$ and only 3 *contact* profiles (with 139 1-Hz measurements) had maximum below-cloud $N_a$ < 200 cm$^{-3}$. A threshold of 300 cm$^{-3}$ used to define a "high-$N_a$ boundary

layer" and cloud microphysical properties and above/below-cloud $N_a$ were compared between 22 C-H and 13 S-H profiles and between 6 C-L and 20 S-L profiles (Table 5). Within low-$N_a$ boundary layers, C-L profiles had slightly lower below-cloud $N_a$ (differences of 1.3 to 26.5 cm$^{-3}$) and similar $N_c$ (insignificant differences) compared to S-L profiles. All other comparisons between the four regimes were consistent with the discussion in subsection 4.3, where a threshold of below-cloud $N_a$ = 350 cm$^{-3}$ was used to define a "high-$N_a$ boundary layer". When the threshold was increased to 400 cm$^{-3}$ and 450 cm$^{-3}$, the qualitative results were unchanged, and C-H (and C-L) profiles had significantly higher $N_c$ and lower $R_e$ compared to S-H (and S-L) profiles. Additionally, there were minor differences between C-H and C-L profiles and between S-H and S-L profiles for these thresholds. Thus, the findings are robust at relates to the choice of this threshold.

*Data Availability.* All ORACLES 2016 in-situ data used in this study are publicly available at https://doi.org/10.5067/Suborbital/ORACLES/P3/2016_V2 (ORACLES Science Team, 2020). This is a fixed-revision subset of the entire ORACLES mission dataset. It contains only the file revisions that were available on 27 May 2020.

*Author Contributions.* SG and GMM conceived the study design and analysis. JRO, DJD and SG processed the in-situ cloud probes data. SG analyzed the data with inputs from GMM, JRO and MRP. GMM, JR and MRP acquired funding. SG, GMM, JRO, DJD, AD, JRP, JR, SEL, MSR and KP collected data on board the NASA P-3. SG wrote the paper with reviews from co-authors.

*Competing interests.* The authors declare that they have no conflict of interest.

*Acknowledgements.* The authors wish to acknowledge Yohei Shinozuka for compiling the merged instrument data files and the entire ORACLES science team for valuable discussions during data acquisition and analysis. We thank Rei Ueyama, Leonhard Pfister and Ju-Mee Ryoo for creating figures obtained from https://bocachica.arc.nasa.gov/ORACLES/. We would like to thank the NASA Ames Earth Science Project Office and the NASA P-3 flight and maintenance crew for the successful deployment. The authors gratefully acknowledge the NOAA Air Resources Laboratory (ARL) for the provision of the HYSPLIT transport and dispersion model and READY website (https://www.ready.noaa.gov) used in this publication. Some of the computing for this project was performed at the OU Supercomputing Center for Education & Research (OSCER) at the University of Oklahoma (OU).

*Financial support.* Funding for this project was obtained from NASA Award #80NSSC18K0222. ORACLES is funded by NASA Earth Venture Suborbital-2 grant NNH13ZDA001N-EVS2. Siddhant Gupta was supported by NASA headquarters under the NASA Earth and Space Science Fellowship grants NNX15AF93G and NNX16A018H and by 80NSSC18K0222.

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

**Table 1: The main parameter used, sampling frequency and measurement range for in-situ instruments installed on the P-3 research aircraft and used within this study.**

| Instrument | Parameter used | Sampling Frequency | Measurement Range | Reference |
|---|---|---|---|---|
| **Rosemount 102** | Temperature | 1 Hz | Nominally -50° to 50°C | Rosemount, Incorporated |
| **Rosemount MADT 2014** | Pressure | 1 Hz | Nominally 30 - 1300 mb | Rosemount, Incorporated |
| **EdgeTech 137 Chilled-Mirror Hygrometer** | Dew Point Temperature | 1 Hz | Nominally -40° to 60°C | EdgeTech Instruments |
| **Global Positioning System** | Latitude, Longitude, Altitude | 1 Hz | -90 to 90° -180 to 180° | |
| **CO/CO2/H2O Analyzer** | CO, $H_2O$ (v) | 1 Hz | 5 to 50,000 ppb, 100 ppm to 100% humidity | Los Gatos Research |
| **CAS** | Droplet n(D) | 10 Hz | 0.5 - 50 µm | Baumgardner et al. (2001) |
| **2D-S** | Droplet Images, asynchronous n(D) | | Nominally 10 - 1,280 µm | Lawson et al. (2006) |
| **HVPS-3** | Droplet Images, asynchronous n(D) | | Nominally 150 - 19,200 µm | Lawson et al. (1998) |
| **King Hot-wire** | Bulk LWC | 25 Hz | 0.05 - 3 g m$^{-3}$ | King et al. (1978) |
| **PCASP** | Aerosol n(D) | 10 Hz | 0.1 - 3 µm | Strapp et al. (1992) |
| **SP2** | Aerosol Absorption | 1 Hz | 55 - 524 nm | Stephens et al. (2003) |
| **HR-ToF-AMS** | Aerosol Mass | 0.2 Hz | 50 - 700 nm | Drewnick et al. (2005) |

**Table 2: List of research flights analyzed with the number of cloud profiles flown and total time spent profiling clouds during each flight. The number of profiles during sawtooth maneuvers is reported within parentheses. The number of profiles and the corresponding sampling time is reported for Contact and Separated profiles during each flight.**

| Flight | Sawtooth + Individual Profiles | Cloud Time | Contact Profiles | Separated Profiles |
|---|---|---|---|---|
| PRF5: September 06 | 4 (4, 5, 4, 6) + 5 | 1327 s | 13 (857 s) | 11 (470 s) |
| PRF7: September 10 | 1 (2) + 7 | 461 s | 0 (0 s) | 9 (461 s) |
| PRF8: September 12 | 1 (6) + 2 | 504 s | 1 (32 s) | 7 (472 s) |
| PRF9: September 14 | 0 (0) + 8 | 574 s | 0 (0 s) | 8 (574 s) |
| PRF11: September 20 | 1 (7) + 6 | 669 s | 13 (669 s) | 0 (0 s) |
| PRF13: September 25 | 2 (2, 3) + 4 | 511 s | 3 (148 s) | 6 (363 s) |
| Total | 9 (39) + 32 | 1h 7m 26s | 30 (1706 s) | 41 (2340 s) |

**Table 3: The total ($OA + SO_4^{2+} + NH_4^+ + NO_3^-$) and OA $M_a$, PCASP $N_a$, and rBC and CO concentrations sampled up to 100 m below cloud base and 100 m above cloud top during four sawtooth maneuvers (S1-S4) flown on 6 September 2016. These values correspond to averages across the individual profiles flown during S1-S4. AOD was sampled during constant altitude flight legs and corresponds to the atmospheric column above the aircraft.**

| Parameter | Location | S1 | S2 | S3 | S4 |
|---|---|---|---|---|---|
| Total $M_a$ ($\mu g\ m^{-3}$) | Above-cloud | 3.4 | 22.9 | 21.7 | 0.8 |
| | Below-cloud | 4.5 | 5.9 | 5.7 | 1.4 |
| OA $M_a$ ($\mu g\ m^{-3}$) | Above-cloud | 2.0 | 16.9 | 13.2 | 0.4 |
| | Below-cloud | 1.9 | 3.5 | 3.4 | 1.0 |
| PCASP $N_a$ ($cm^{-3}$) | Above-cloud | 241 | 151 | 1334 | 16 |
| | Below-cloud | 354 | 5 | 390 | 72 |
| | | | 327 | | |
| rBC ($cm^{-3}$) | Above-cloud | 66 | 516 | 700 | 10 |
| | Below-cloud | 72 | 111 | 130 | Not available |
| CO (ppb) | Above-cloud | 95 | 196 | 230 | 96 |
| | Below-cloud | 93 | 103 | 117 | 88 |
| AOD | Above-cloud | 0.33 | 0.37 | 0.49 | 0.39 |

**Table 4: The range of time, latitude, longitude, above-cloud AOD and cloud top height ($Z_T$) for cloud profiles flown during the six flights. The lowest altitude where above-cloud $N_a > 500\ cm^{-3}$ occurred during the flight ($Z_{500}$) is in the far-right column.**

| Date | Time (UTC) | Latitude (°S) | Longitude (°E) | AOD | $Z_T$ (m) | $Z_{500}$ (m) |
|---|---|---|---|---|---|---|
| September 6 | 08:46 - 12:35 | 10.2 - 19.7 | 9.0 - 11.9 | 0.27 - 0.49 | 359 - 1002 | 680 |
| September 10 | 09:09 - 12:36 | 14.1 - 18.7 | 4.0 - 8.6 | 0.21 - 0.29 | 990 - 1201 | 1800 |
| September 12 | 11:16 - 12:26 | 9.7 - 12.9 | -0.3 - 3.0 | 0.25 - 0.29 | 1146 - 1226 | 1200 |
| September 14 | 09:36 - 14:16 | 16.4 - 18.1 | 7.5 - 9.0 | 0.31 - 0.32 | 635 - 824 | 2350 |
| September 20 | 08:44 - 13:11 | 15.7 - 17.3 | 8.9 - 10.5 | 0.42 - 0.56 | 432 - 636 | 600 |
| September 25 | 10:59 - 13:51 | 10.9 - 14.3 | 0.8 - 4.3 | 0.27 - 0.38 | 729 - 1124 | 1170 |

Table 5: Aerosol and cloud properties were averaged across all *contact/separated* profiles flown in low $N_a$ and high $N_a$ boundary layers. These averages were compared between *contact* and *separated* profiles. The values listed below represent the 95% confidence intervals (from a two-sample t-test) when the differences were statistically significant. Positive values indicate the average for *contact* profiles was higher and "insignificant" denotes the differences were statistically insignificant.

| Maximum below-cloud $N_a$ (cm$^{-3}$) | Below-cloud $N_a$ (cm$^{-3}$) | Above-cloud $N_a$ (cm$^{-3}$) | $N_c$ (cm$^{-3}$) | $R_e$ (μm) | LWC (g m$^{-3}$) |
|---|---|---|---|---|---|
| Low $N_a$ (< 300 cm$^{-3}$) | -1.3 - -26.5 | 498.0 - 565.5 | insignificant | -0.1 - -0.6 | insignificant |
| High $N_a$ (> 300 cm$^{-3}$) | 48.3 - 78.2 | 746.7 - 884.3 | 80.8 - 92.8 | -1.1 - -1.3 | 0.0 – 0.02 |
| Low $N_a$ (< 350 cm$^{-3}$) | insignificant | 592.7 - 669.4 | 22.8 - 34.9 | -0.3 - -0.9 | insignificant |
| High $N_a$ (> 350 cm$^{-3}$) | 39.1 - 70.5 | 737.8 - 884.4 | 75.5 - 88.4 | -1.2 - -1.6 | 0.0 - 0.02 |

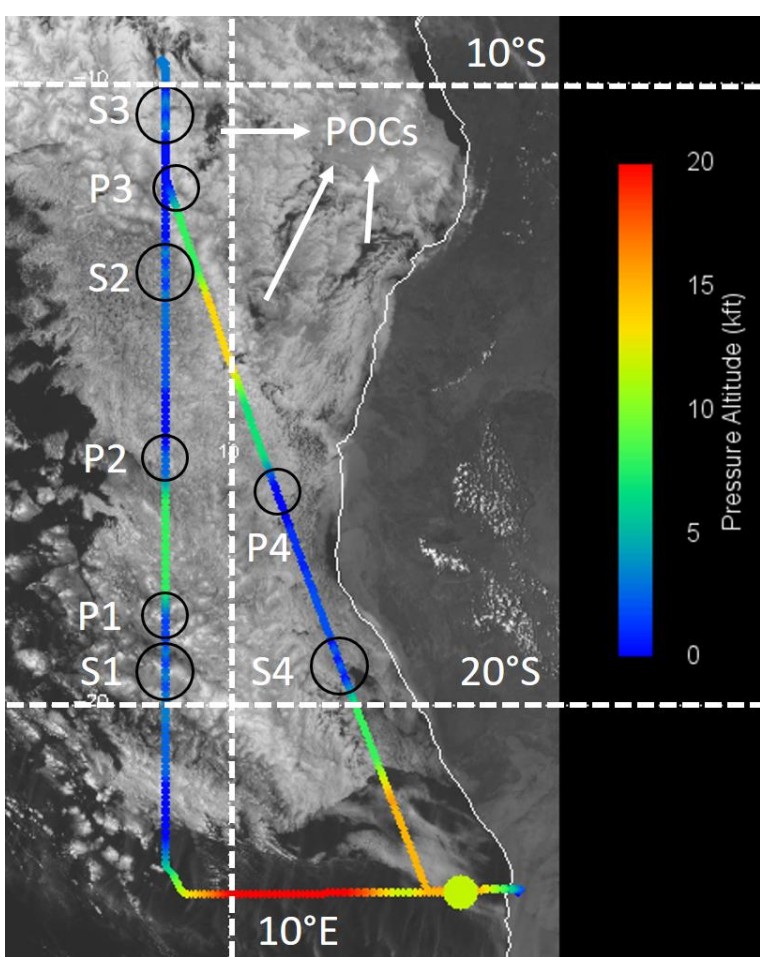

Figure 1: Visible image from the Spinning Enhanced Visible and Infrared Imager at 14:00 UTC on 6 September 2016 (PRF5), overlaid by the PRF5 flight track, and colored by flight altitude. Circles indicate sawtooth maneuver (S) and individual cloud profile (P) locations (https://bocachica.arc.nasa.gov/ORACLES/).

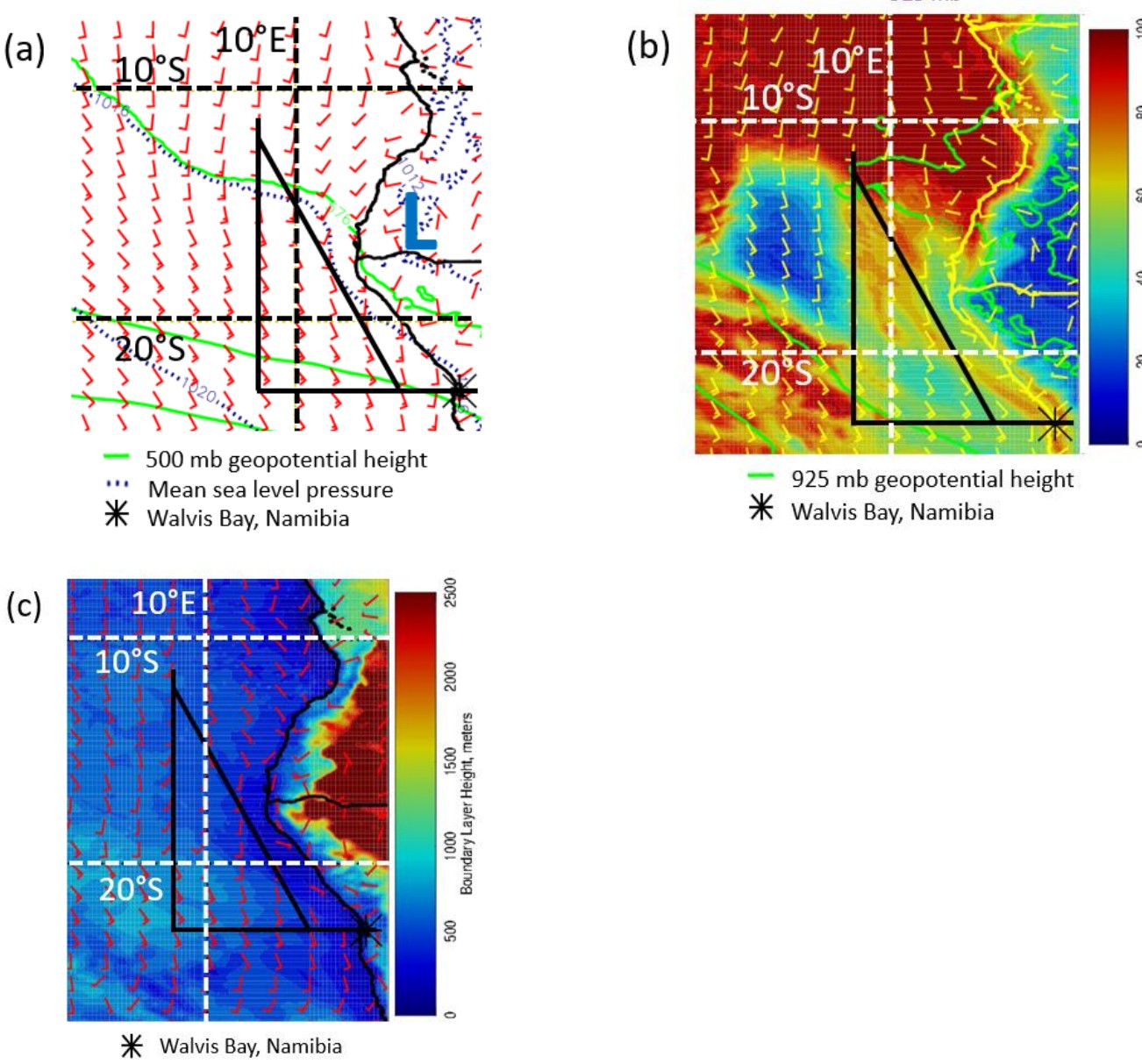

**Figure 2: 0-hour European Centre for Medium-Range Weather Forecasts reanalysis at 12:00 UTC on 6 September 2016 for (a) mean sea level pressure, 500mb geopotential height and surface wind, (b) 925mb relative humidity, geopotential height and wind, and (c) boundary layer height and 900 mb wind (https://bocachica.arc.nasa.gov/ORACLES/).**

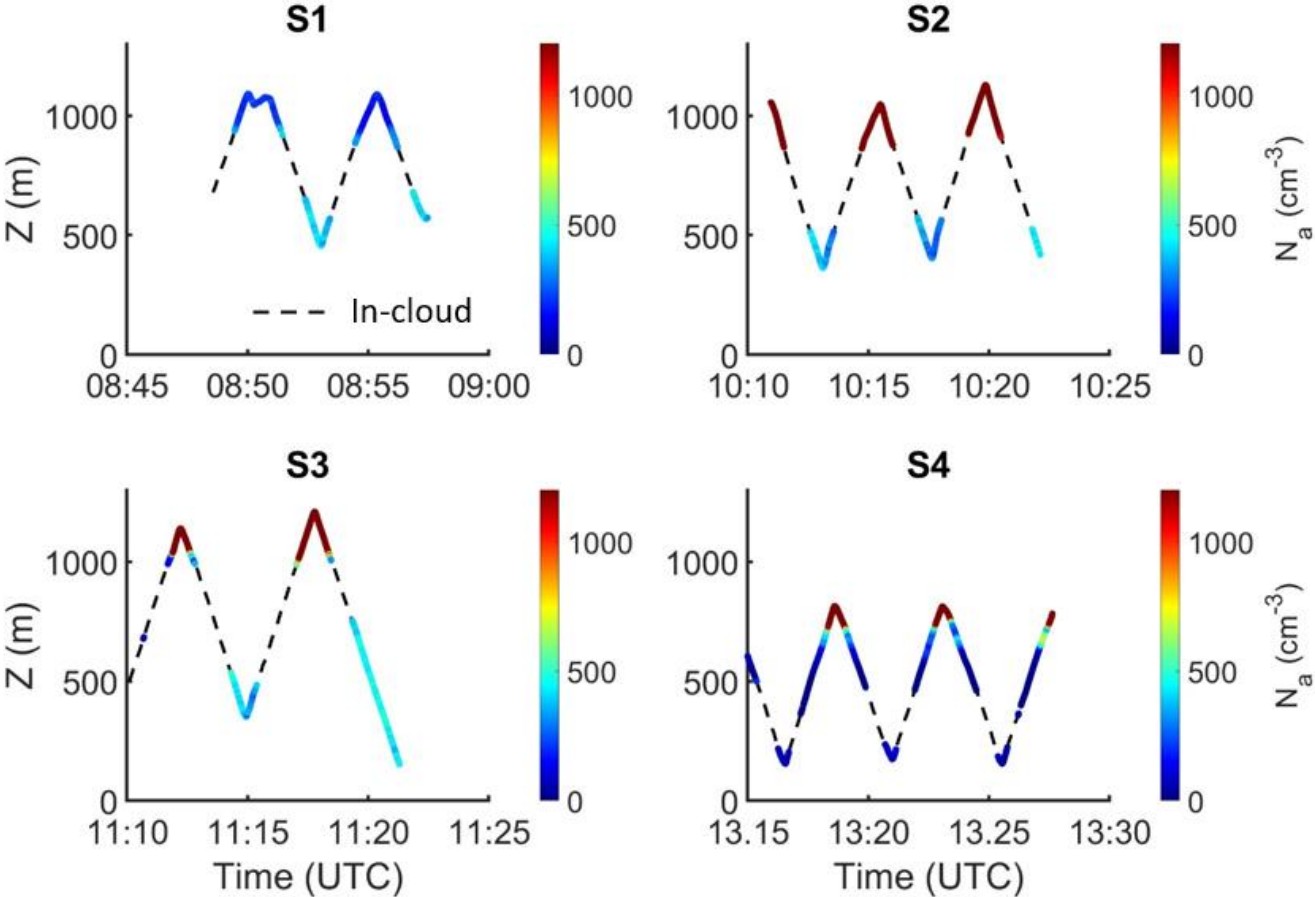

**Figure 3: P-3 aircraft altitude as a function of time, colored by PCASP accumulation-mode (0.1 < D < 3 μm) $N_a$ for 4 sawtooth maneuvers flown on 6 September 2016. In-cloud $N_a$ are masked due to potential for droplet shattering on the PCASP probe inlet.**

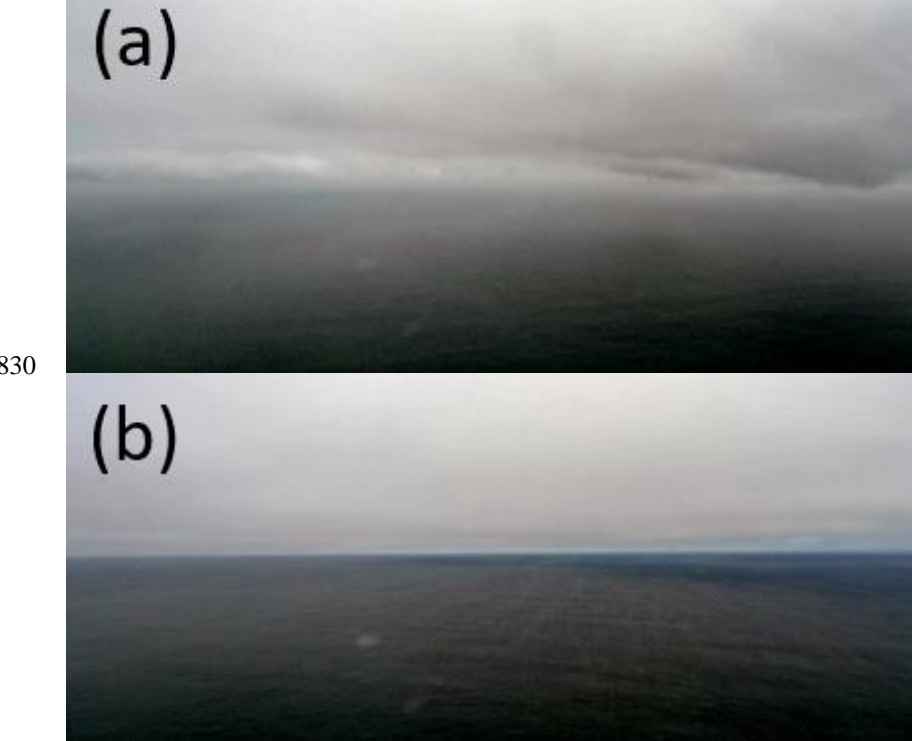

**Figure 4: Snapshots of the boundary layer sampled below (a) S1 showing shallow cumulus and stratocumulus layers with varying bases, and (b) S4 showing stratocumulus clouds with a uniform base (NSRC/NASA Airborne Science Program)**

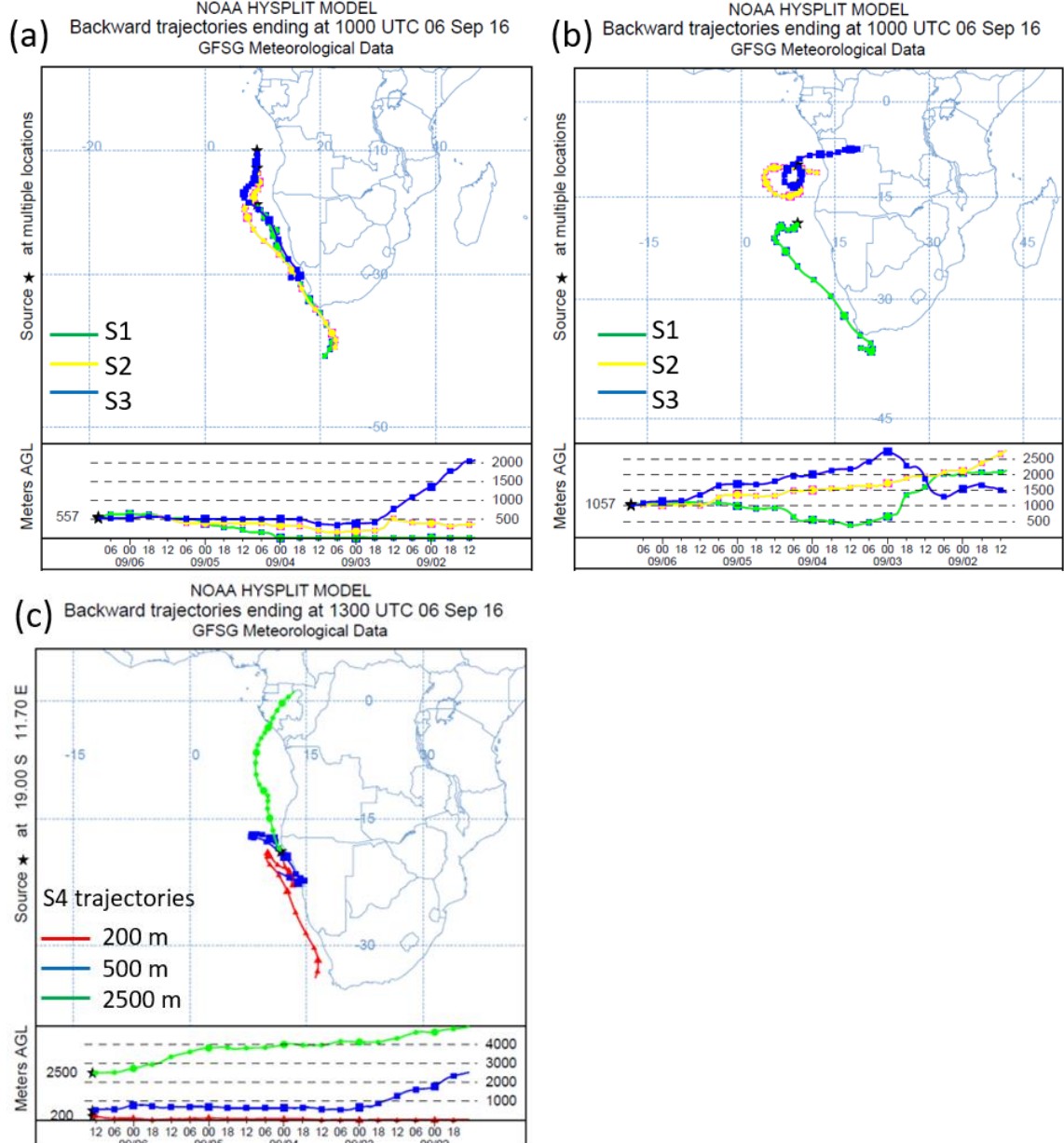

**Figure 5: 5-day back-trajectories from the Hybrid Single Particle Lagrangian Integrated Trajectory model for sawtooth maneuvers flown on 6 September 2016 (a) ending at 10:00 UTC for S1-S3 at 500 m, (b) ending at 10:00 UTC for S1-S3 at 1000 m and (c) ending at 13:00 UTC for S4 at 200 m, 500 m and 2500 m (altitudes represent values above mean sea level)**

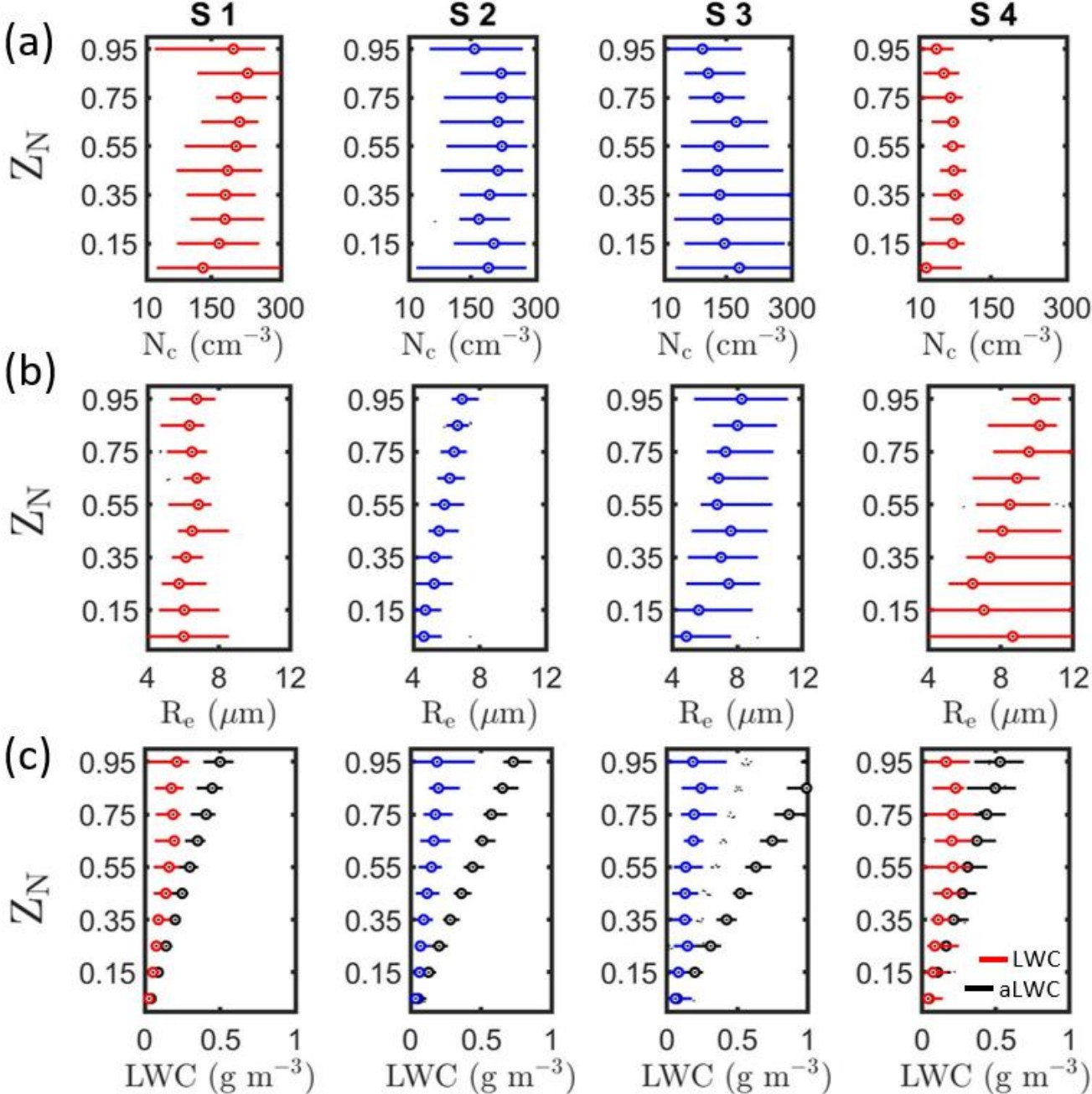

Figure 6: Vertical profiles of (a) $N_c$, (b) $R_e$ and (c) LWC and aLWC as a function of $Z_N$ for the 4 sawtooth maneuvers. Maneuvers with contact (separation) between the biomass-burning aerosol layer and cloud tops shown in blue (red).

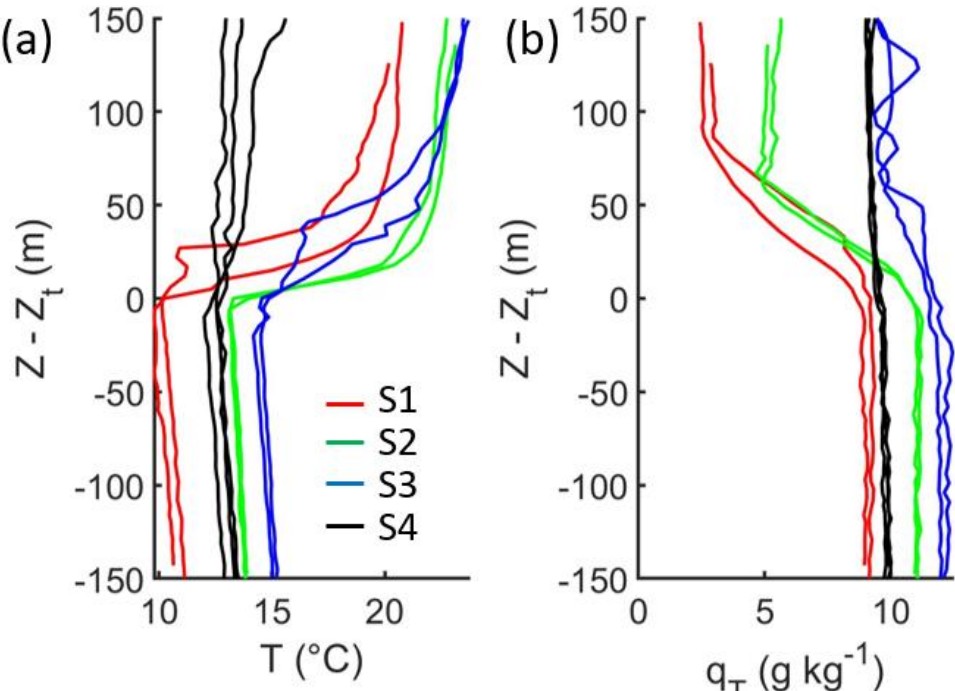

Figure 7: Vertical profiles of (a) T and (b) $q_T$ as a function of distance from cloud top. Each line corresponds to an individual ascent through cloud during a sawtooth. The profiles flown during S2 and S3 (S1 and S4) had contact (separation) between the above-cloud biomass-burning aerosol layer and cloud tops.

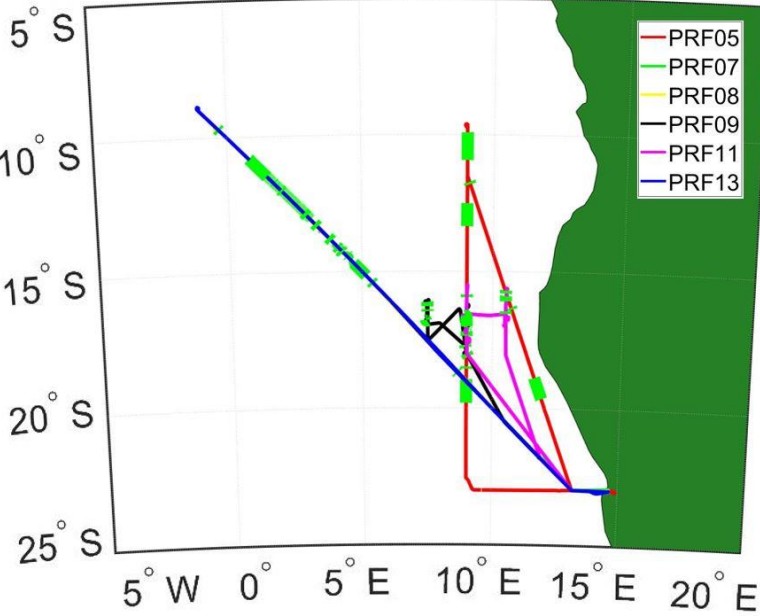

Figure 8: Flight tracks from PRFs 5, 7, 8, 9, 11 and 12 flown on 6, 10, 12, 14, 20 and 25 September 2016 with green segments indicating location of cloud profiles (flight tracks from PRFs 7 and 8 coincide with PRF13 and hence are not visible).

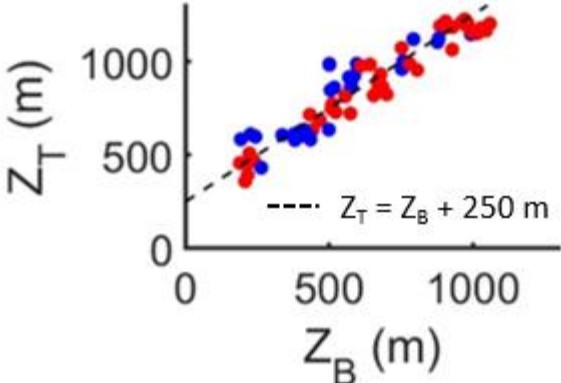

**Figure 9: Cloud base and top heights for contact (blue) and separated (red) profiles flown during the six PRFs.**

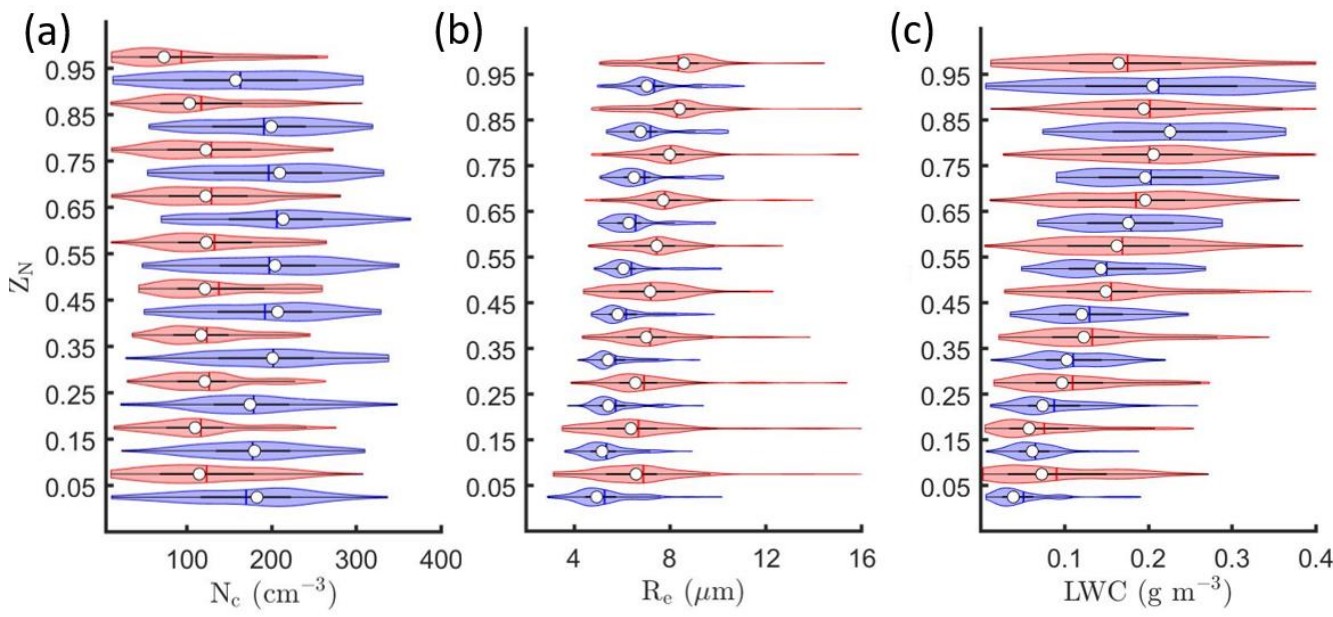

**Figure 10: Kernel density estimates (indicated by the width of shaded area) and boxplots showing the 25th (Q1), 50th (white point) and 75th (Q3) percentile for (a) $N_c$ (b) $R_e$ and (c) LWC as a function of $Z_N$ for contact (blue) and separated (red) profiles.**

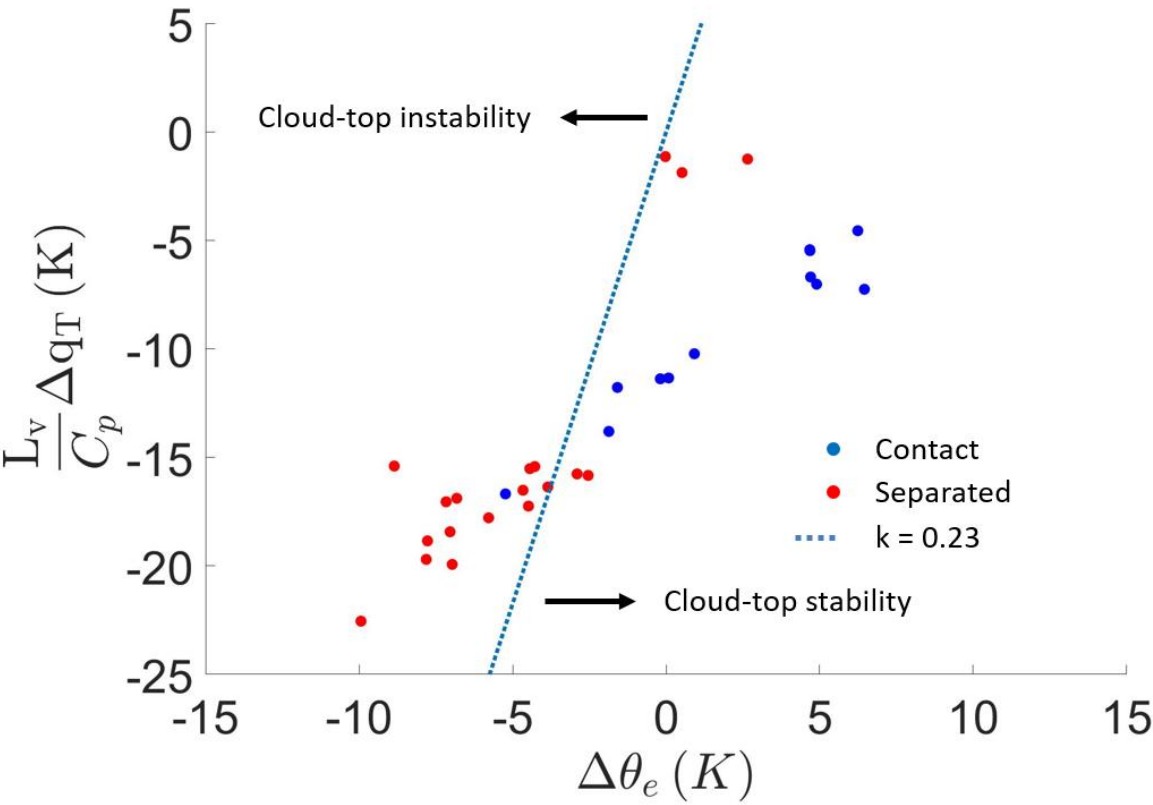

**Figure 11: Difference between equivalent potential temperature (θe) and total water mixing ratio (qT) measured within cloud and 100 m above cloud top for contact (blue) and separated (red) profiles (only ascents through cloud shown).**

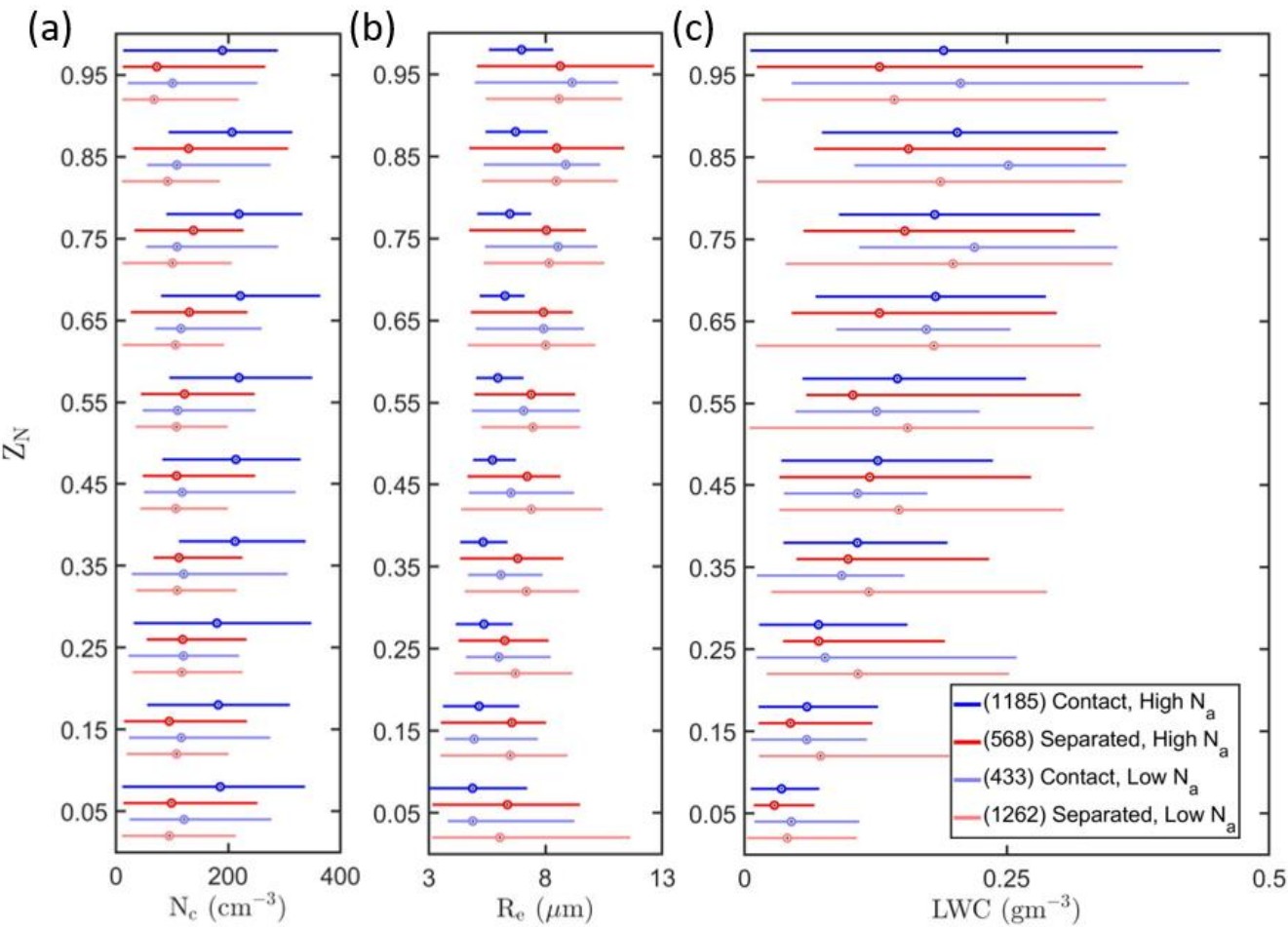

**Figure 12: Boxplots representing vertical profiles of (a) $N_c$, (b) $R_e$, and (c) LWC as a function of $Z_N$ for contact (blue) and separated (red) profiles within boundary layers with high $N_a$ (> 350 cm$^{-3}$) (darker) or low $N_a$ (< 350 cm$^{-3}$) (lighter). The number of 1 Hz measurements within each regime is listed within parentheses.**