# Peer review of "Impact of the Variability in Vertical Separation between Biomass-Burning Aerosols and Marine Stratocumulus on Cloud Microphysical Properties over the Southeast Atlantic"

_Atmospheric Chemistry and Physics, 2020_

## Referee Comment (RC1) · Jonathan Taylor (Referee) · 9 Dec 2020

Review of "Impact of the Variability in Vertical Separation between Biomass-Burning Aerosols and Marine Stratocumulus on Cloud Microphysical Properties over the Southeast Atlantic" by Gupta et al

This paper presents new and interesting measurements of aerosols and clouds from the ORACLES field study in this special issue. The measurements appear to be of a high quality and are presented well (in terms of the graphs)- the author has taken a large dataset and condensed it down into some useful figures. The first half of the paper is excellent- I have little of substance to say on the introduction and experimental sections. Then I reached the results sections 3&4, where the reader is presented with a monumental wall of text, which ends up being quite difficult to read. I was waiting for a discussion to help me make sense of it all, how it relates to the indirect and semi-direct effects, and which aspects are the most important, and then the discussion never came! If the intention is simply to provide numbers to put in a model, then I think you should rebrand this as a measurement report. If the intention is to give some original scientific insight using your results, then you should do this by adding a discussion section.

**Major comments**

Please add some sort of graphical or tabular summary of your results/conclusions. A bit like Table 5, but with words to help the reader.

L408 – 413 "The differences between contact and separated profiles in low-Na…" and "Consequently, the differences between contact and separated profiles in high-Na…."
These two statements are your actual science conclusions. Everything prior to this is largely a stamp collecting exercise. As in, we know from previous literature and your introduction that when pollution plumes mix into the cloud layer, the clouds become more polluted, and most of your paper is about putting some numbers on that. These two statements where you are summarising what you have inferred from these numbers about what are the main processes and drivers, those are actual conclusions. The first part of your discussion should focus on how you have come to these conclusions. The second should relate your results back to what you discussed in the introduction. The radiative effects depend on COT and microphysical properties at cloud top. You could discuss how the clean clouds with low below-cloud $N_a$ are more susceptible to addition of extra aerosol than the clouds with a boundary layer that is already polluted (Twomey, 1991). And throughout your discussion, include some comparisons to relevant literature- look at other papers in the special issue, as well as other studies (certainly VOCALS, possibly DACCIWA and others).

Figure 9: This is actually a really key figure. It shows that the cloud depth is pretty much constant for all clouds sampled. If that wasn't the case then you couldn't do your normalised height plots, and you would have concerns that the semi-direct effects could dominate. Please add in somewhere about how much you think the semi-direct effects might have affected your results

**Minor comments**

L102 the end of the introduction is quite abrupt- it would be good to have a couple of sentences outlining what you do in this paper, such as what the different sections of the results are. In this study we present results from ORACLES. First we show a case study from one flight, then we look at a statistical analysis of several flights together.

2 Instrumentation section- there's some strange details in here. For example, I think I understand what you mean by the PCASP gain stage correction but I don't see how it's relevant if all you're doing is taking the total concentration. You mention lots of different cloud probes, hotwire probes etc., but then you only use the CAS and the 2DS for the cloud measurements. Also why do you mention the gas analyser first, when the aerosol and cloud are the focus of your measurements?

L114 Using the PCASP for total aerosol concentration- do you have an idea of the size distribution and what fraction of aerosol might be below the lower cutoff diameter of the PCASP?

L125 Whichever cloud probes you end up using, please briefly state how the size was calibrated, and give an estimate of the uncertainty in size and concentration

151 Is the CAS better than the CDP? I would normally think the CDP is better, but that's just using our instruments, yours may be different

Sections 3&4 Please divide each of these up into several subsections to break it up, and to guide the reader by summarising what you are talking about in each section

Figure 3: I suggest you make these just profiles, the time information isn't particularly useful. Also mark on cloud base and top heights with dashed lines

L119 the part about the big decrease in $N_c$ between $0 - 0.25$ in $Z_n$. This seems to only be one bin, so is it just a blip? It's difficult to tell how much data you have in any of these bins

L230 what part of the profile are these ratios from? The average?

L257 It is strange to mention these 4 regimes here and then not explain what they are.

L281 These violin plots- are they figure 10? If so then reference it here.

L296 "Buoyancy and humidity…." This is so weird and out of place. It would work much better if you start a new subsection with the next paragraph, and put it somewhere in that subsection.

L300-365 I have little to comment other than this is so densely written, it is very difficult to pick anything out as a reader. What I did was I looked at your plots, and I asked why Figure 12 only has profiles of $N_c$ and not of Re and LWC as well?

The thresholds of 300 vs 350 thing, I think that makes it more confusing. You seem to come out with similar conclusions regardless of which number you use, right? So I think just pick one. On a different year or different time of the year, the particle concentrations might be different anyway so the number you pick is somewhat arbitrary. This is especially true when you have Appendix A which is all about your choice of threshold anyway.

Figure 12 At cloud base the Contact, high $N_a$ numbers are significantly higher than the separated, high $N_a$ numbers. How much of the differences you see are due to differences in below-cloud $N_a$ versus mixing in from above? You would expect the cloud base $N_c$ to be driven by the below-cloud $N_a$. Is the below-cloud $N_a$ similar for both sets of cases? And how much does this affect the other differences between the contact and separated profiles?

Table 1: Please remove any instruments you haven't used in your analysis

Table 3: In the caption, state that the insitu measurements only cover up to the max altitude on the profile, whereas the AOD cover the whole of the above-cloud column

Table 5: What does "---" mean? No data? Or not statistically significant? Or something else?

Figures 6 & 10 Make sure the Y axes go from 0 to 1. Also figure 10 please plot these side by side

**Technical corrections**

L43 Hartmann et al- do you have a more recent reference?

L68 absorption increasing buoyancy- isn't this the semi-direct effect?

L68 It's a bit unclear what the next sentence is trying to say- are you trying to say that as particle-laden air is entrained into the cloud, this increases Nc but also can decrease LWC, depending on the humidity of the air that is mixing in?

L83 You haven't yet defined ORACLES in the main text

L182 "bulk LWC > 10g m-3" please check/correct

L203 "$N_a$ < 500cm-3" Please check/correct, should it be >500?

L226 "drizzle concentration decreased near cloud base which led to the decrease in median $R_e$" Does it not increase near cloud base?

L244 Do you mean higher below-cloud $N_a$, rather than above? In general you need to be careful talking about above-cloud $N_a$, because your AOD measurements suggest all profiles had high above-cloud $N_a$ if you go high enough

L250 "Higher $N_c$ in the cloud layer…" This is a confusing sentence. How about "As the high-$N_a$ air from the free troposphere entrains through the inversion, $N_c$ in the top of the cloud layer increased. This change provides evidence for the aerosol indirect effect". Having said that…does it provide evidence of the indirect effect? The indirect effect being the radiative part, not just the microphysics.

L273 What is P1? Profile 1 obviously….but you have not explained your naming convention.

**Reference**

Twomey, S. (1991). Aerosols, clouds and radiation. *Atmospheric Environment Part A, General Topics*, 25(11), 2435–2442. https://doi.org/10.1016/0960-1686(91)90159-5

Finally, thankyou it has been interesting to read. I've not seen someone dig into such fine detail in something as basic as profiles before!

---

## Referee Comment (RC2) · Anonymous Referee #2 · 11 Dec 2020

General:

This is a very well done article, and the conclusions are reasonable.

Specific comment: Line 200, there needs to be a space after the 5 (... 5 to 10-minute ...) not a - sign. This is very insignificant, really.
* * *

---

## Referee Comment (RC3) · Anonymous Referee #3 · 12 Dec 2020

Review of "Impact of the Variability in Vertical Separation between Biomass-Burning Aerosols and Marine Stratocumulus on Cloud Microphysical Properties over the Southeast Atlantic" by Gupta et al.

This study reports on the important issue of smoke-cloud interactions with a focus on the vertical separation of aerosol layers from cloud tops, which is of importance in the southeast Atlantic Ocean region where there can be large smoke plumes above and within the boundary layer. This study makes use of ORACLES data, specifically from six research flights. Statistics are provided about the number of cases where the

aerosol layers (> 500 cm-3) are within 100 m of cloud tops ( "contact") or in excess of 100 m from tops (called "separated"). Subsequently, cloud properties and free tropospheric humidity are compared for these two categories of cases. A finding was that droplet evaporation (from entrainment drying at cloud top) was enhanced in cases where plumes were above 100 m from cloud tops (called "separated"); this was coincident with greater reductions in cloud drop number concentration and liquid water content near cloud tops. Another finding was that sub-cloud aerosol number concentrations were typically higher for "contact" cases (> 350 cm-3). Also, the "contact" cases with high aerosol concentrations in the boundary layer had higher drop concentrations as compared to "separated" cases. The paper was well written and easy to follow. At least one of the tables was difficult for me to digest but in general the tables and figures were clear too. The results are important and I am fully supportive of publication after the comments below are addressed.

Specific Comments:

Table 5: Took me several time to read the caption to try to understand the table and I am still not sure I understand it.

Line 76-85: I suspect it may be important to refer to this study somewhere here or elsewhere in the paper owing to its high relevance:

Rajapakshe, C., et al. 2017. Seasonally transported aerosol layers over southeast Atlantic are closer to underlying clouds than previously reported. Geophysical Research Letters, 44, 5818–5825. https://doi.org/10.1002/2017GL073559

Line 118-119: Give a brief description of how the collection efficiency was computed and handled for the data presented.

Line 182: Are the authors sure they mean LWC > 10 g m-3? That seems too high (by 2 orders of magnitude).

Throughout the paper I suggest the authors consult with 3 other recent references to

at least mention them for the sake of comparison and contrast. The Mardi et al. (2018) paper quantifies in detail smoke layer separation from stratocumulus cloud top heights, while their 2019 paper digs into cloud-smoke interactions that are related to results from this study. The Diamond et al. (2018) examines smoke-cloud interactions too over the same region as that of this study. In particular I find that the threshold to use for what constitutes a smoke plume (i.e., its base altitude) to be quite important, for which results of studies like this can be sensitive to; I found it interesting that the criteria in this study seemed to be Na > 500 cm-3, whereas that in the Mardi et al. papers was 1000 cm-3.

References:

Mardi, A.H., et al. 2019. Effects of Biomass Burning on Stratocumulus Droplet Characteristics, Drizzle Rate, and Composition. J Geophys Res-Atmos 124, 12301-12318.

Mardi, A.H., et al. 2018. Biomass Burning Plumes in the Vicinity of the California Coast: Airborne Characterization of Physicochemical Properties, Heating Rates, and Spatiotemporal Features. J Geophys Res-Atmos 123, 13560-13582.

Diamond, M. S., et al. 2018. Time-dependent entrainment of smoke presents an observational challenge for assessing aerosol-cloud interactions over the southeast Atlantic Ocean. Atmospheric Chemistry and Physics, 18(19), 14623–14636. https://doi.org/10.5194/acp-18-14623-2018

Line 374-375: Are the authors sure they have unambiguous evidence of these causal relationships? This is always a tricky thing with aircraft data and I caution the authors to reconsider if they want to use this strong language.
* * *

---

## Author Comment (AC1) · 12 Feb 2021

The authors would like to thank Jonathan Taylor for their time and efforts in reviewing the manuscript. This document contains author responses to reviewer comments. Reviewer comments are in red and author responses are in black. Removed text is in "*quotes and italicized*" and added/replacement text is in "**quotes and in bold**".

This paper presents new and interesting measurements of aerosols and clouds from the ORACLES field study in this special issue. The measurements appear to be of a high quality and are presented well (in terms of the graphs)- the author has taken a large dataset and condensed it down into some useful figures. The first half of the paper is excellent- I have little of substance to say on the introduction and experimental sections. Then I reached the results sections 3&4, where the reader is presented with a monumental wall of text, which ends up being quite difficult to read. I was waiting for a discussion to help me make sense of it all, how it relates to the indirect and semi-direct effects, and which aspects are the most important, and then the discussion never came! If the intention is simply to provide numbers to put in a model, then I think you should rebrand this as a measurement report. If the intention is to give some original scientific insight using your results, then you should do this by adding a discussion section.

The authors thank Jonathan Taylor for the very thorough review. The comments provide valuable inputs to improve the manuscript. A discussion was added in section 5 to provide more context and directions for future work.

**Major comments**

Please add some sort of graphical or tabular summary of your results/conclusions. A bit like Table 5, but with words to help the reader.

Table 5 was updated to include comparisons of liquid water content (LWC) between the four regimes. Figure 12 was updated to include vertical profiles of $R_e$ and LWC. The numbers from Table 5 are discussed in subsection 4.4 and Appendix – A.

L408 – 413 "The differences between contact and separated profiles in low-Na…" and "Consequently, the differences between contact and separated profiles in high-Na…."
These two statements are your actual science conclusions. Everything prior to this is largely a stamp collecting exercise. As in, we know from previous literature and your introduction that when pollution plumes mix into the cloud layer, the clouds become more polluted, and most of your paper is about putting some numbers on that. These two statements where you are summarising what you have inferred from these numbers about what are the main processes

and drivers, those are actual conclusions. The first part of your discussion should focus on how you have come to these conclusions. The second should relate your results back to what you discussed in the introduction. The radiative effects depend on COT and microphysical properties at cloud top. You could discuss how the clean clouds with low below-cloud Na are more susceptible to addition of extra aerosol than the clouds with a boundary layer that is already polluted (Twomey, 1991). And throughout your discussion, include some comparisons to relevant literature- look at other papers in the special issue, as well as other studies (certainly VOCALS, possibly DACCIWA and others).

The following changes were made to include relevant results from previous literature in addition to the discussion in section 5 and not including specific changes in response to comments below:

The following text was added after Line 299 in the old manuscript:
"**Recent studies have shown there is strong correlation between above-cloud AOD and water vapor within air masses originating from the African continent (Deaconu et al., 2019; Pistone et al., 2021). Longwave cooling by water vapor within the BBA layer leads to decreased cloud-top cooling and cloud-top dynamics are influenced by distinct radiative contributions from water vapor and absorbing aerosols.**"
The following text was added after Line 323 in the old manuscript:
"**This is consistent with significantly higher average H (267 m) for *contact* profiles compared to *separated* profiles (213 m). Braun et al. (2018) found a negative correlation between H and adiabaticity (ratio of the measured and the adiabatic liquid water path) which is consistent with *contact* profiles having lower LWC/aLWC and higher H compared to *separated* profiles.**"

Figure 9: This is actually a really key figure. It shows that the cloud depth is pretty much constant for all clouds sampled. If that wasn't the case then you couldn't do your normalised height plots, and you would have concerns that the semi-direct effects could dominate. Please add in somewhere about how much you think the semi-direct effects might have affected your results

Modelling studies have found that shortwave absorption by absorbing aerosols above clouds leads to increased cloud water due to greater stability and entrainment suppression (Johnson et al., 2004; Sakaeda et al., 2011). However, minor differences in LWC between *contact* and *separated* profiles suggest the response of cloud water to the absorption component of the semidirect effect likely did not vary between these profiles. It is noted the modelling studies did not consider microphysical interactions between the aerosol and cloud layers which can be affected by the radiative contribution of water vapor within the aerosol layer and changes in buoyancy at cloud tops (Deaconu et al., 2019; Herbert et al., 2020).

Wilcox (2010) used satellite observations to show the semi-direct effect can impact stratocumulus clouds over the southeast Atlantic. The authors believe the semidirect effect likely affected both *contact* and *separated* profiles since a layer of absorbing aerosols was always located above the sampled clouds (above-cloud AOD > 0.2 was retrieved for all profiles). Since the LWC differences were limited to the top 20 % of the cloud layer, it is likely the differences between these profiles were primarily driven by cloud-top entrainment, evaporative cooling, and buoyancy reversal. We are reluctant to speculate on the impact of the semidirect effect on our conclusions since that would require modelling efforts to examine the relative microphysical impacts of the semidirect and indirect effects.

**Minor comments**

L102 the end of the introduction is quite abrupt- it would be good to have a couple of sentences outlining what you do in this paper, such as what the different sections of the results are. In this study we present results from ORACLES. First we show a case study from one flight, then we look at a statistical analysis of several flights together.

The following text was added at the end of the section 1:
"**The remainder of the paper is organized as follows. The instrumentation used in the analysis is described in Section 2 along with the procedures for processing the data. A case study of the 6 September 2016 research flight is presented in Section 3. The meteorological and aerosol conditions present are examined and profiles of $N_c$, $R_e$, and LWC are compared for four sawtooth maneuvers flown at locations where clouds were in contact and separated from above-cloud BBA. In Section 4, measurements from six research flights are analysed to investigate buoyancy associated with cloud-top evaporative cooling and profiles of $N_c$, $R_e$, and LWC are compared for boundary layers with similar and varying aerosol loading. Finally, the conclusions and their impact on the understanding of aerosol-cloud interactions are discussed in Section 5.**"

2 Instrumentation section- there's some strange details in here. For example, I think I understand what you mean by the PCASP gain stage correction but I don't see how it's relevant if all you're doing is taking the total concentration. You mention lots of different cloud probes, hotwire probes etc., but then you only use the CAS and the 2DS for the cloud measurements. Also why do you mention the gas analyser first, when the aerosol and cloud are the focus of your measurements?

The PCASP gain stage correction was required to calculate the aerosol size distribution because the higher voltages on the gain stages resulted in increased rejection of particles, particularly

within the first 5 size bins of the instrument. The increased particle rejection resulted in an undercounting of aerosols, and thus lower total aerosol concentration. The total aerosol concentration was calculated by integrating the aerosol number concentration within each size bin within the accumulation mode size range (0.1 to 3 $\mu$m).

The authors have left the mention of the entire suite of cloud probes deployed during the ORACLES field campaign within the text for the following purposes:

1. This is the first manuscript from the ORACLES field campaign with primary focus on data collected by the in-situ cloud probes.
2. We hope this manuscript will serve as a reference for the cloud probe data quality and processing procedures relating to the ORACLES 2016 deployment.

Nevertheless, rows containing information about unused instrumentation were removed from Table 1 for brevity.

Line 107 in the old manuscript was changed.

"*…with in-situ probes for sampling meteorological conditions, aerosols and clouds (Table 1), among other instrumentation.*" To

"**…with in-situ probes for sampling aerosols, clouds and meteorological conditions (Table 1), among other instrumentation.**"

Lines 108-113 in the old manuscript describing the gas analyzer were moved to the end of the instrumentation section.

Line 118 in the old manuscript ending "*…was corrected.*" Was changed to "**…was corrected to calculate the total aerosol concentration.**"

L114 Using the PCASP for total aerosol concentration- do you have an idea of the size distribution and what fraction of aerosol might be below the lower cutoff diameter of the PCASP?

The total aerosol concentration ($N_a$) was calculated using the PCASP size distribution since the focus of this study is on cloud microphysical properties which are primarily influenced by accumulation-mode aerosols acting as cloud condensation nuclei (CCN). The impact of "small-diameter particles" (aerosols smaller than 0.1 $\mu$m in diameter) on cloud properties was likely to be limited since these particles will be unable to activate due to their low hygroscopicity (Che et al., 2021). This assumption is consistent with previous studies that found strong correlations between CCN at 0.2 % supersaturation and PCASP $N_a$ (Mardi et al., 2019).

Smaller particles were sampled by the Ultra-High Sensitivity Aerosol Spectrometer (UHSAS). The PCASP n(D) peaked at about 0.3 $\mu$m while the UHSAS n(D) peaked at about 0.18 $\mu$m. For aerosols greater than 0.18 $\mu$m in diameter, the UHSAS n(D) underestimated the PCASP n(D). For the case

study on 6 September 2016, the average UHSAS $N_a$ was 23.5 cm$^{-3}$ while the average PCASP $N_a$ was 823.9 cm$^{-3}$. Therefore, the fraction of aerosols below the PCASP cutoff diameter was low which would further limit their impact on cloud microphysical properties and the results presented in the manuscript.

L125 Whichever cloud probes you end up using, please briefly state how the size was calibrated, and give an estimate of the uncertainty in size and concentration

The following text was added:
"**The in-situ probes used here (CAS, 2D-S, HVPS-3, and PCASP) were calibrated by the manufacturers prior to and shortly after the deployment. During the deployment, performance checks according to the instrument manuals were completed to determine any change in instrument performance. This included monitoring the CAS and 2D-S voltages and temperatures during flights and passing calibration particles through the CAS sample volume to determine any change in the relationship between particle size and peak signal voltage.**"

Estimates of uncertainties in sizing and concentrations are addressed by Baumgardner et al. (2017).
The following change was made:
"*Baumgardner et al. (2017) discuss the general operating characteristics of the in-situ cloud probes…*" was
"**Baumgardner et al. (2017) discuss the general operating characteristics and measurement uncertainties of the in-situ cloud probes…**"

151 Is the CAS better than the CDP? I would normally think the CDP is better, but that's just using our instruments, yours may be different

CAS data were used since CDP data were unusable for the entire deployment due to an optical misalignment issue.

Sections 3&4 Please divide each of these up into several subsections to break it up, and to guide the reader by summarising what you are talking about in each section

Sections 3 and 4 have been divided into subsections within the updated manuscript.

Figure 3: I suggest you make these just profiles, the time information isn't particularly useful. Also mark on cloud base and top heights with dashed lines

The authors believe the time-height plots in Fig. 3 illustrate the aircraft altitude during sawtooth maneuvers while also illustrating the location and concentration of above- and below-cloud aerosols. In our view, cloud base and top heights ($Z_B$ and $Z_T$) are sufficiently discernible by the dashed lines along the altitude profile. To add horizontal lines for each profile would make the figure cluttered since $Z_B$ and $Z_T$ varied between the profiles during each maneuver.

L119 the part about the big decrease in Nc between 0 – 0.25 in Zn. This seems to only be one bin, so is it just a blip? It's difficult to tell how much data you have in any of these bins

We believe this comment refers to Line 219 instead of line 119.
Each $Z_N$ bin for S2 and S3 contained between 30 to 35 1-Hz cloud samples. This could be a blip for S2 but looks to be a robust trend for S3. Furthermore, there is a distinct difference between the trends in median $N_c$ and $R_e$ near cloud base between contact and separated profiles from the case study. Contact profiles (S2 and S3) had decreasing $N_c$ and increasing $R_e$ as $Z_N$ increased from 0.05 to 0.25 but separated profiles (S1 and S4) had increasing $N_c$ and decreasing $R_e$ over these levels. However, these trends were more subtle within Fig. 12 which suggests these trends were likely specific to the four sawtooth maneuvers from the case study.

L230 what part of the profile are these ratios from? The average?

The LWC/aLWC ratios were determined by averaging the values over the cloud layer.
Line 230 has been changed to mention this detail:
"*Lower LWC/aLWC for S2 and S3…*" was changed to "**Lower LWC/aLWC (averaged over the cloud layer) for S2 and S3…**"

L257 It is strange to mention these 4 regimes here and then not explain what they are.

The line is edited to directly describe the regime classification criteria and keep the regime definitions within Section 4.
"*61 profiles were further classified into four new regimes based on below-cloud $N_a$ to quantify the differences in $N_c$ between contact and separated profiles within boundary layers with similar below-cloud $N_a$.*" was changed to
"**61 *contact* and *separated* profiles were further classified as low-$N_a$ or high-$N_a$ profiles based on the below-cloud $N_a$. This was done to quantify the differences in $N_c$ and $R_e$ between *contact* and *separated* profiles within boundary layers with similar below-cloud $N_a$**"

L281 These violin plots- are they figure 10? If so then reference it here.

*"…were examined using violin plots…"* was changed to **"…were examined in Fig. 10 using violin plots…"**

L296 "Buoyancy and humidity…." This is so weird and out of place. It would work much better if you start a new subsection with the next paragraph, and put it somewhere in that subsection.

The line was moved to the next paragraph which is now under the sub-section titled "Cloud-top Evaporative Cooling".

L300-365 I have little to comment other than this is so densely written, it is very difficult to pick anything out as a reader. What I did was I looked at your plots, and I asked why Figure 12 only has profiles of Nc and not of Re and LWC as well?

Section 4 has been divided into sub sections to guide the reader.

Figure 12 was updated to include profiles of $R_e$ and LWC for a below-cloud $N_a$ threshold of 350 cm$^{-3}$. The $N_c$ profile for the below-cloud $N_a$ threshold of 300 cm$^{-3}$ was removed and the Fig. 12 caption was updated:
*"Boxplots representing $N_c$ as a function of $Z_N$ for contact (blue) and separated (red) profiles within boundary layers with high-$N_a$ (darker) or low-$N_a$ (lighter). The number of 1 Hz measurements within each regime is listed within parentheses. A high-$N_a$ boundary layer is defined as having maximum $N_a$ up to 100 m below cloud base (a) greater than 300 cm$^{-3}$ and (b) greater than 350 cm$^{-3}$"* to
**"Boxplots representing vertical profiles of (a) $N_c$, (b) $R_e$, and (c) LWC as a function of $Z_N$ for contact (blue) and separated (red) profiles within boundary layers with high $N_a$ (> 350 cm$^{-3}$) (darker) or low $N_a$ (< 350 cm$^{-3}$) (lighter). The number of 1 Hz measurements within each regime is listed within parentheses."**

The thresholds of 300 vs 350 thing, I think that makes it more confusing. You seem to come out with similar conclusions regardless of which number you use, right? So I think just pick one. On a different year or different time of the year, the particle concentrations might be different anyway so the number you pick is somewhat arbitrary. This is especially true when you have Appendix A which is all about your choice of threshold anyway.

The discussion, when a threshold of 300 cm$^{-3}$ was used, has been moved to Appendix-A.

Figure 12 At cloud base the Contact, high Na numbers are significantly higher than the separated, high Na numbers. How much of the differences you see are due to differences in below-cloud Na

versus mixing in from above? You would expect the cloud base Nc to be driven by the below-cloud Na. Is the below-cloud Na similar for both sets of cases? And how much does this affect the other differences between the contact and separated profiles?

The below-cloud $N_a$ had statistically insignificant differences between Contact-low $N_a$ and Separated-low $N_a$ profiles. Therefore, it is unlikely the differences in $N_c$ for these profiles were driven solely by below-cloud $N_a$. Between Contact-high $N_a$ and Separated-high $N_a$ profiles, there was a significant difference in below-cloud $N_a$, but it was lower than the corresponding difference in average $N_c$. Therefore, below-cloud $N_a$ alone would be insufficient to explain the microphysical changes.

Table 1: Please remove any instruments you haven't used in your analysis

The caption for Table 1 was changed.
"*Primary measurement, sampling frequency and measurement range of the in-situ instruments installed on the P-3 research aircraft*" to
"**The main parameter used, sampling frequency and measurement range for in-situ instruments installed on the P-3 research aircraft and used within this study.**".

The following instruments were removed from Table 1 along with the corresponding references (unless cited outside Table 1):
Turbulent Air Motion Measurement System (TAMMS), Cloud Droplet Probe (CDP), Phase Doppler Interferometer (PDI), Cloud Imaging Probe (CIP), CAPS Hot wire (LWC 100), and Ultra High Sensitivity Aerosol Spectrometer (UHSAS).

Table 3: In the caption, state that the insitu measurements only cover up to the max altitude on the profile, whereas the AOD cover the whole of the above-cloud column

The caption for Table 3 was changed.
 "*The above- and below-cloud aerosol and trace gas concentrations with the above-cloud Aerosol Optical Depth (AOD) for four sawtooth maneuvers (S1-S4) flown on 6 September 2016. The values correspond to averages across the individual profiles flown during each sawtooth maneuver.*" to
"**The total (OA + $SO_4^{2+}$ + $NH_4^+$ + $NO_3^-$) and OA $M_a$, PCASP $N_a$, and rBC and CO concentrations sampled up to 100 m below cloud base and 100 m above cloud top during four sawtooth maneuvers (S1-S4) flown on 6 September 2016. These values correspond to averages across the individual profiles flown during S1-S4. AOD was sampled during constant altitude flight legs and corresponds to the atmospheric column above the aircraft.**"

Table 5: What does "---" mean? No data? Or not statistically significant? Or something else?

The table contains the 95 % confidence intervals for differences between variable means. The "---" was meant to denote the differences which were statistically insignificant. Therefore, "---" was changed to "**insignificant**" for clarity.

Figures 6 & 10 Make sure the Y axes go from 0 to 1. Also figure 10 please plot these side by side

Figure 10 was edited to have the panels next to each other. However, we have decided to keep the original axis labels.
Line 212 within the old manuscript states "…the bin with $0 < Z_N < 0.1$ (represented by the midpoint, $Z_N = 0.05$) included data collected over the bottom 10% of the cloud layer". We have kept the original terminology where a $Z_N$ bin is referred to using the bin midpoint. Therefore, it would be appropriate to have y-axis labels to represent the same values as the text.

**Technical corrections**

L43 Hartmann et al- do you have a more recent reference?

The following changes were made to the first paragraph to add more recent references:
"*Globally averaged annual cloud cover can reach up to 61% of the Earth's surface (Warren et al., 1988) and contributes a radiative forcing of about -22 W m$^{-2}$ to Earth's energy budget (Hartmann et al., 1992)*" has been changed to
"**Clouds cover about two-thirds of the Earth's surface (Stubenrauch et al., 2013) and exert a global net cloud radiative effect (CRE) of about – 17.1 W m$^{-2}$ on Earth's energy budget (Loeb et al., 2009).**"

The following sentence was changed from "*Cloud radiative effects (CREs) include…*" to "**The net CRE includes …**".
Line 52 in the old manuscript was moved to the previous paragraph and changed:
"*General Circulation Models have large uncertainties in their estimates of CREs and the associated cloud feedbacks, in part due to their treatment of low-level clouds, particularly stratocumulus (Boucher et al., 2013)*" to
"**General Circulation Models have large uncertainties and inter-model spread in estimates of the net CRE (Boucher et al., 2013). This is partly due to strong underestimation of the subtropical marine stratocumulus cloud cover and the associated CRE (Wang and Su, 2013).**"

L68 absorption increasing buoyancy- isn't this the semi-direct effect?

This sentence was moved to the following paragraph starting "BBA over the southeast Atlantic…" where the semidirect aerosol effect was discussed.

L68 It's a bit unclear what the next sentence is trying to say- are you trying to say that as particle-laden air is entrained into the cloud, this increases Nc but also can decrease LWC, depending on the humidity of the air that is mixing in?

This paragraph describes how the local moisture profile and cloud-top entrainment can modulate the impact of aerosol-cloud interactions on cloud properties. This sentence discusses the findings from the cited literature that cloud-top entrainment leads to lower LWC in clouds with higher $N_c$ and precipitation suppression due to the decrease in droplet sizes. LWC in such polluted clouds with higher $N_c$ would decrease due to cloud top entrainment unless the overlying air was humid.

L83 You haven't yet defined ORACLES in the main text

The sentence was edited, and the acronym was defined here:
"**Observations from the NASA ObseRvations of Aerosols above CLouds and their intEractionS (ORACLES) field campaign found…**"

The acronym definition on Line 99 was removed:
"*The ObseRvations of Aerosols above Clouds and their intEractionS (ORACLES) field campaign provides a unique dataset…*" was changed to "**The ORACLES field campaign provides a unique dataset…**"

L182 "bulk LWC > 10g m-3" please check/correct

"*LWC > 10*" has been corrected to "**LWC > 0.05**" to reflect the correct value for the LWC threshold used.

L203 "Na < 500cm-3" Please check/correct, should it be >500?

This is left unchanged. Values of "$N_a < 500$ cm$^{-3}$" were reported up to at least 200 m above cloud tops during S1 (Fig. 3).

L226 "drizzle concentration decreased near cloud base which led to the decrease in median Re" Does it not increase near cloud base?

The authors meant the drizzle concentration decreased with $Z_N$ near cloud base. The following change has been made for better clarity.

"*For S4, drizzle concentration decreased near cloud base which led to the decrease in median $R_e$.*" is changed to "**For S4, drizzle concentration decreased from $Z_N$ = 0.05 to 0.25 which led to the decrease in median $R_e$ over these heights.**"

L244 Do you mean higher below-cloud Na, rather than above? In general you need to be careful talking about above-cloud Na, because your AOD measurements suggest all profiles had high above-cloud Na if you go high enough

The sentence is left unchanged as the authors meant "above-cloud $N_a$" with "below-cloud $N_a$" mentioned in the following sentence.

The reviewer suggestion about relating AOD with above-cloud $N_a$ is noted. To address this, the authors refer to the average aerosol concentration sampled up to 100 m above cloud tops whenever "above-cloud $N_a$" is mentioned or compared between cloud profiles.

This was done under the assumption that aerosol concentrations beyond 100 m above cloud top had little impact on cloud properties (in terms of the indirect effect). This assumption was tested in the appendix by using a range of vertical distances between the aerosol and cloud layers to define "separation".

L250 "Higher Nc in the cloud layer…" This is a confusing sentence. How about "As the high-Na air from the free troposphere entrains through the inversion, Nc in the top of the cloud layer increased. This change provides evidence for the aerosol indirect effect". Having said that…does it provide evidence of the indirect effect? The indirect effect being the radiative part, not just the microphysics.

Line 250 was changed to include a brief discussion on cloud optical thickness:

"*Higher $N_c$ in the cloud layer due to entrainment mixing of free tropospheric air with significantly higher above-cloud $N_a$ provided evidence of the aerosol indirect effect due to the presence of above-cloud BBA*" was changed to

"**These microphysical changes would also impact cloud reflectance (Twomey, 1991) as seen by the significantly higher cloud optical thickness ($\tau$) of *contact* profiles compared to *separated* profiles (differences of 2.5 to 8.2). The increase in t and the cloud reflectance provides observational evidence of the aerosol indirect effect over the southeast Atlantic due to contact between above-cloud BBA and the stratocumulus clouds.**"

L273 What is P1? Profile 1 obviously….but you have not explained your naming convention.

"*P1 had…*" was changed to "**Profile 1 (P1) had…**"

**Reference**

Twomey, S. (1991). Aerosols, clouds and radiation. *Atmospheric Environment Part A, General Topics*, *25*(11), 2435–2442. https://doi.org/10.1016/0960-1686(91)90159-5

Finally, thankyou it has been interesting to read. I've not seen someone dig into such fine detail in something as basic as profiles before!

The authors made the following changes to the manuscript in addition to reviewer comments:

1. Citations for Adebiyi and Zuidema (2016) and Wilcox (2010) were added.
2. "*BL*" was changed to "**boundary layer**" throughout the manuscript.
- Lines 167, 172, 199, 246, captions for Fig. 2 and 4
3. "*Figure*" was abbreviated as "**Fig.**"
- Lines 161, 163, 166, 167, 169, 171, 172, 177, 178, 183, 186, 191, 193, 194, 204, 212, 216, 217, 224, 228, 233, 261, 277, 284, 289, 292, 313, 318, 341, and 351.
4. Minor formatting changes were made to Table 2 and 3 for clarity.
5. A legend was added in Fig. 3 to represent in-cloud altitudes.

REFERENCES:
Che, H., Stier, P., Gordon, H., Watson-Parris, D., and Deaconu, L.: Cloud adjustments dominate the overall negative aerosol radiative effects of biomass burning aerosols in UKESM1 climate model simulations over the south-eastern Atlantic, Atmos. Chem. Phys., 21, 17–33, https://doi.org/10.5194/acp-21-17-2021, 2021.
Deaconu, L. T., Ferlay, N., Waquet, F., Peers, F., Thieuleux, F., and Goloub, P.: Satellite inference of water vapour and above-cloud aerosol combined effect on radiative budget and cloud-top processes in the southeastern Atlantic Ocean, Atmos. Chem. Phys., 19, 11613–11634, https://doi.org/10.5194/acp-19-11613-2019, 2019.
Herbert, R. J., Bellouin, N., Highwood, E. J., and Hill, A. A.: Diurnal cycle of the semi-direct effect from a persistent absorbing aerosol layer over marine stratocumulus in large-eddy simulations, Atmos. Chem. Phys., 20, 1317–1340, https://doi.org/10.5194/acp-20-1317-2020, 2020.
Johnson, B. T., Shine, K. P., and Forster, P. M.: The semi-direct aerosol effect: Impact of absorbing aerosols on marine stratocumulus, Q. J. R. Meteorol. Soc., 130, 1407–1422, 2004.
Sakaeda, N., Wood, R., and Rasch, P. J.: Direct and semidirect aerosol effects of southern African biomass burning aerosol, J. Geophys. Res., 116, D12205, doi:10.1029/2010JD015540, 2011.

Wilcox, E. M.: Stratocumulus cloud thickening beneath layers of absorbing smoke aerosol, Atmos. Chem. Phys., 10, 11769–11777, https://doi.org/10.5194/acp-10-11769-2010, 2010.

---

## Author Comment (AC2) · 12 Feb 2021

Reviewer Comment: "General: This is a very well done article, and the conclusions are reasonable."

Author response: "The authors would like to thank the reviewer for their comments."

Reviewer Comment: "Specific comment: Line 200, there needs to be a space after the 5 (... 5 to 10-minute...) not a - sign. This is very insignificant, really."

Author response: "Thank you, this was corrected."
* * *

---

## Author Comment (AC3) · 12 Feb 2021

The authors would like to thank the anonymous reviewer for their time and efforts in reviewing the manuscript. This document contains author responses to reviewer comments. Reviewer comments are in red and author responses are in black. Removed text is in "*quotes and italicized*" and added/replacement text is in "**quotes and in bold**".

Review of "Impact of the Variability in Vertical Separation between Biomass-Burning Aerosols and Marine Stratocumulus on Cloud Microphysical Properties over the Southeast Atlantic" by Gupta et al.

This study reports on the important issue of smoke-cloud interactions with a focus on the vertical separation of aerosol layers from cloud tops, which is of importance in the southeast Atlantic Ocean region where there can be large smoke plumes above and within the boundary layer. This study makes use of ORACLES data, specifically from six research flights. Statistics are provided about the number of cases where theaerosol layers (> 500 cm-3) are within 100 m of cloud tops ( "contact") or in excess of 100 m from tops (called "separated"). Subsequently, cloud properties and free tropospheric humidity are compared for these two categories of cases. A finding was that droplet evaporation (from entrainment drying at cloud top) was enhanced in cases where plumes were above 100 m from cloud tops (called "separated"); this was coincident with greater reductions in cloud drop number concentration and liquid water content near cloud tops. Another finding was that sub-cloud aerosol number concentrations were typically higher for "contact" cases (> 350 cm-3). Also, the "contact" cases with high aerosol concentrations in the boundary layer had higher drop concentrations as compared to "separated" cases. The paper was well written and easy to follow. At least one of the tables was difficult for me to digest but in general the tables and figures were clear too. The results are important and I am fully supportive of publication after the comments below are addressed.

The authors thank the reviewer for their comments and support for publication. Specific reviewer comments are addressed below.

Specific Comments:
Table 5: Took me several time to read the caption to try to understand the table and I am still not sure I understand it.

The caption for Table 5 was changed for clarity.
"*Differences between the average below- and above-cloud $N_a$, and the average $N_c$ and $R_e$ measured in the cloud layer for contact profiles relative to separated profiles. The differences are classified by the maximum below-cloud $N_a$ within the boundary layer and correspond to 95% confidence intervals based on a two-sample t-test (not reported when p > 0.05).*" to

"**Aerosol and cloud properties were averaged across all *contact/separated* profiles flown in low N$_a$ and high N$_a$ boundary layers. These averages were compared between *contact* and *separated* profiles. The values listed below represent the 95% confidence intervals (from a two-sample t-test) when the differences were statistically significant. Positive values indicate the average for *contact* profiles was higher and "insignificant" denotes the differences were statistically insignificant.**"

Line 76-85: I suspect it may be important to refer to this study somewhere here or elsewhere in the paper owing to its high relevance:
Rajapakshe, C., et al. 2017. Seasonally transported aerosol layers over southeast Atlantic are closer to underlying clouds than previously reported. Geophysical Research Letters, 44, 5818–5825. https://doi.org/10.1002/2017GL073559

The authors note the relevance of this study here. The following text was added in section 1:
"**Rajapakshe et al. (2017) found the aerosol layer was located within 360 m above the cloud layer for about 60% of the Cloud-Aerosol Transport System (CATS) lidar night-time scenes over the southeast Atlantic.**"

Line 118-119: Give a brief description of how the collection efficiency was computed and handled for the data presented.

A time- and composition-dependent collection efficiency (CE) was applied to correct for the incomplete vaporization of mixed phase particles. As discussed in Middlebrook et al., (2012), the CE is primarily determined by the efficiency with which a particle's impaction upon the vaporizer is detected. This is turn is mostly explained by the phase of the particle, with liquid aerosol collected more efficiently than neutralized aerosol because it is less likely to bounce off the heater and escape detection. Liquid aerosol is primarily acidic, and the acidity of the free-tropospheric aerosol is assessed by comparing the molar ratio of NH$_4$ to 2xSO$_4$. The use of the NH$_4$/(2SO$_4$) ratio is a simplification of the NH$_{4,measured}$/NH$_{4,predicted}$ relationship. NH$_{4,predicted}$ is the amount of ammonium required to neutralize the inorganic anions observed by the AMS. The collection efficiency is then determined from *CE=max(0.5, 1- NH$_4$/(2xSO$_4$))*, with a value of 0.5 serving as the lower limit, consistent with the default value applied within most field campaigns (Middlebrook et al., 2012).

The following text was added after line 119:
"**A time- and composition-dependent collection efficiency (CE) was applied to AMS data. The molar ratio of ammonium to sulphate (NH$_4$/(2xSO$_4$)) was calculated to assess the acidity of liquid aerosol which are collected more efficiently compared to neutralized aerosol. Thus, CE**"

was determined as the maximum between 0.5 and (1- NH$_4$/(2xSO$_4$)), with a value of 0.5 serving as the lower limit, consistent with estimates from most previous field campaigns (Middlebrook et al., 2012)."

Middlebrook, A. M., Bahreini, R., Jimenez, J. L., and Canagaratna, M. R.: Evaluation of composition-dependent collection efficiencies for the aerodyne aerosol mass spectrometer using field data, Aerosol Sci. Technol., 46, 258–271, doi:10.1080/02786826.2011.620041s, 2012.

Line 182: Are the authors sure they mean LWC > 10 g m-3? That seems too high (by 2 orders of magnitude).

"*LWC > 10*" has been corrected to "**LWC > 0.05**" to reflect the correct value for the LWC threshold used.

Throughout the paper I suggest the authors consult with 3 other recent references toat least mention them for the sake of comparison and contrast. The Mardi et al. (2018) paper quantifies in detail smoke layer separation from stratocumulus cloud top heights, while their 2019 paper digs into cloud-smoke interactions that are related to results from this study. The Diamond et al. (2018) examines smoke-cloud interactions too over the same region as that of this study. In particular I find that the threshold to use for what constitutes a smoke plume (i.e., its base altitude) to be quite important, for which results of studies like this can be sensitive to; I found it interesting that the criteria in this study seemed to be Na > 500 cm-3, whereas that in the Mardi et al. papers was 1000 cm-3.

The authors note the importance of referencing these studies and comparing their observations with the results presented here. These studies are referenced at appropriate points within the manuscript. In addition to discussions within subsection 4.4 and section 5, the following additions were made:

The following text was added after Line 199 of the old manuscript:
"**This is also likely to be associated with the history of entrainment mixing of polluted free tropospheric air into the boundary layer prior to these observations (Diamond et al., 2018).**"
The following text was added after Line 208 of the old manuscript:
"**In a previous study, a significantly higher threshold (PCASP N$_a$ = 1000 cm$^{-3}$) was used to identify the BBA layer above stratocumulus clouds off the coast of California (Mardi et al., 2018). The sensitivity of the threshold chosen in this study is examined in Appendix-A and using a threshold of 1000 cm$^{-3}$ would have no significant impact on the results presented in this study.**"

The following text was added within subsection 4.4:

"**Previous studies have argued the changes in $N_c$ due to the impact of BBA are more strongly correlated with below-cloud $N_a$ compared to above-cloud $N_a$ (Diamond et al., 2018; Mardi et al., 2019). However, these results suggest that although the differences in $N_c$ were lower than the differences in above-cloud $N_a$, significant changes in $N_c$ and $R_e$ were associated with contact with above-cloud BBA, and these changes were independent of the below-cloud aerosol loading.**"

References:

Mardi, A.H., et al. 2019. Effects of Biomass Burning on Stratocumulus Droplet Characteristics, Drizzle Rate, and Composition. J Geophys Res-Atmos 124, 12301-12318.

Mardi, A.H., et al. 2018. Biomass Burning Plumes in the Vicinity of the California Coast: Airborne Characterization of Physicochemical Properties, Heating Rates, and Spatiotemporal Features. J Geophys Res-Atmos 123, 13560-13582.

Diamond, M. S., et al. 2018. Time-dependent entrainment of smoke presents an observational challenge for assessing aerosol-cloud interactions over the southeast Atlantic Ocean. Atmospheric Chemistry and Physics, 18(19), 14623–14636. https://doi.org/10.5194/acp-18-14623-2018

Line 374-375: Are the authors sure they have unambiguous evidence of these causal relationships? This is always a tricky thing with aircraft data and I caution the authors to reconsider if they want to use this strong language.

The authors acknowledge the caveats presented by aircraft data. These are snapshots in space and time and may not reflect the conditions of the entire domain.

The sentence has been moved to follow the next sentence starting "In-situ measurements" was changed:

"*The presence of biomass-burning aerosols immediately above cloud tops impacts $N_c$, $R_e$, and LWC through cloud-top entrainment and increases the free tropospheric temperature and humidity*"
to
"**These observations suggest the presence of biomass-burning aerosols immediately above cloud tops was associated with changes in vertical profiles of $N_c$, $R_e$, and LWC due to cloud-top entrainment and increases in the free tropospheric temperature and humidity.**"

The authors made the following changes to the manuscript in addition to reviewer comments:

1. Citations for Adebiyi and Zuidema (2016) and Wilcox (2010) were added.
2. "*BL*" was changed to "**boundary layer**" throughout the manuscript.
- Lines 167, 172, 199, 246, captions for Fig. 2 and 4
3. "*Figure*" was abbreviated as "**Fig.**"
- Lines 161, 163, 166, 167, 169, 171, 172, 177, 178, 183, 186, 191, 193, 194, 204, 212, 216, 217, 224, 228, 233, 261, 277, 284, 289, 292, 313, 318, 341, and 351.
4. Minor formatting changes were made to Table 2 and 3 for clarity.
5. A legend was added in Fig. 3 to represent in-cloud altitudes.